# Converting endogenous genes of the malaria mosquito into simple non-autonomous gene drives for population replacement

Astrid Hoermann, Sofia Tapanelli, Paolo Capriotti, Giuseppe Del Corsano, Ellen KG Masters, Tibebu Habtewold, George K Christophides, Nikolai Windbichler*

Department of Life Sciences, Imperial College London, London, United Kingdom

**Abstract** Gene drives for mosquito population replacement are promising tools for malaria control. However, there is currently no clear pathway for safely testing such tools in endemic countries. The lack of well-characterized promoters for infection-relevant tissues and regulatory hurdles are further obstacles for their design and use. Here we explore how minimal genetic modifications of endogenous mosquito genes can convert them directly into non-autonomous gene drives without disrupting their expression. We co-opted the native regulatory sequences of three midgut-specific loci of the malaria vector *Anopheles gambiae* to host a prototypical antimalarial molecule and guide-RNAs encoded within artificial introns that support efficient gene drive. We assess the propensity of these modifications to interfere with the development of *Plasmodium falciparum* and their effect on fitness. Because of their inherent simplicity and passive mode of drive such traits could form part of an acceptable testing pathway of gene drives for malaria eradication.

*For correspondence:
nikolai.windbichler@imperial.ac.uk

Competing interests: The authors declare that no competing interests exist.

## Introduction

After more than a decade of sustained success in the fight against malaria, data from 2015 onwards suggests that no significant progress in reducing global malaria cases has been achieved (*World malaria report, 2019*). The rise of insecticide resistance in mosquito vectors and drug resistance in parasites highlights the urgent need to develop new tools if malaria eradication is to remain a viable goal.

Synthetic gene drives spreading by super-Mendelian inheritance within vector populations have been proposed as an area-wide genetic strategy for the control of malaria (*Raban et al., 2020*). They mimic the mechanism of proliferation of a class of naturally occurring selfish genes found in protists, mitochondria, and chloroplasts called homing endonucleases and could be deployed to modify the genetic makeup of disease vector populations (*Burt, 2003*; *Windbichler et al., 2011*). Proof-of-principle laboratory experiments of CRISPR/Cas9-based gene drives for population suppression, which aims at the elimination of the target population (*Kyrou et al., 2018*; *Hammond et al., 2016*), as well as population replacement (*Gantz et al., 2015*; *Pham et al., 2019*; *Adolfi et al., 2020*), which aims to spread an anti-malarial trait within a vector population, suggest that these strategies could be deployed to reduce malaria transmission in the field.

The success of population replacement in particular hinges on the availability of molecules that can efficiently block Plasmodium development within the mosquito vector, as well as ways to express such elements. Few pre-characterized promoter elements driving tissue-specific expression in infection-relevant tissues currently exist in malaria vectors. Examples include the regulatory elements of

the zinc carboxypeptidase A1 (CP), peritrophin1 (Aper1), or the vitellogenin (Vg) genes that have been reported to drive transgene expression (*Nolan et al., 2011*; *Ito et al., 2002*; *Abraham et al., 2005*; *Nirmala et al., 2006*; *Volohonsky et al., 2015*). A variety of endogenous and exogenous effector molecules have been expressed in transgenic mosquitoes to interfere with the development of the parasite including anti-malarial regulators such as Cecropin A (*Kim et al., 2004*), fibrinogen domain-containing immunolectin 9 (FBN9) (*Simões et al., 2017*), TEP1 (*Volohonsky et al., 2017*), the transcription factor Rel2 (*Dong et al., 2011*), as well as the protein kinase Akt (*Corby-Harris et al., 2010*; *Arik et al., 2015*) and the phosphatase and tensin homolog (PTEN) (*Hauck et al., 2013*) involved in insulin signalling. Synthetic peptides Vida3 (*Ito et al., 2002*) and SM1, engineered as a quadruplet (*Meredith et al., 2011*), the bee venom phospholipases A2 (PLA2) (*Abraham et al., 2005*; *Moreira et al., 2002*), as well as single-chain antibodies (scFvs, m1C3, m4B7, and m2A10) targeting the ookinete protein chitinase 1 and the circumsporozoite protein (CSP) (*Isaacs et al., 2011*; *Isaacs et al., 2012* ) have been used. Recently, microRNA sponges have been suggested as modulators of mosquito immunity (*Dong et al., 2020a*). The performance of these effector molecules has been evaluated using laboratory strains of the human parasite *Plasmodium falciparum* or the rodent parasite *Plasmodium berghei*, but the efficacy against circulating polymorphic *P. falciparum* strains of human malaria parasites is currently unknown.

A further consideration is the population dynamics and persistence of transmission-blocking gene drives (*Beaghton et al., 2017*; *Noble et al., 2017*), which would suggest a trade-off between the targeting of a conserved sequence with a resulting fitness cost versus the targeting of a neutral genomic site that shows poor conservation and engenders the evolution of resistance to the drive. A number of ways have been suggested to work around this, for example, the linking of rescue-copies of essential genes to drive elements (*Adolfi et al., 2020*; *Esvelt et al., 2014*; *Champer et al., 2020a*). Cleave and rescue elements are a related approach that does not rely on homing (*Oberhofer et al., 2019*; *Oberhofer et al., 2020*; *Champer et al., 2020b*).

We have recently modelled an alternative approach to achieve population replacement that was specifically designed to address these questions (*Nash et al., 2019*). The approach, we have termed integral gene drive (IGD), imitates the way naturally occurring homing endonuclease genes propagate, that is, by copying themselves into highly conserved sites within genes. They do not disrupt their target genes due to their association with introns or inteins (*Hafez and Hausner, 2012*; *Barzel et al., 2011*). We analysed the theoretical behaviour of IGDs featuring the complete molecular separation of drive and effector functions into minimal modifications of endogenous host genes at two or more different genomic loci (*Nash et al., 2019*). This approach takes advantage of the promoter and also of its surrounding regulatory regions of the modified endogenous locus. To accommodate the guide-module required for subsequent gene drive and a marker-module required for monitoring transgenesis, we suggested to insert an intron into the effector gene.

In the present study we sought to scope the feasibility of this approach in the African malaria mosquito *A. gambiae*. Our aim was to generate minimal genetic modifications of mosquito genes in situ, which would turn particular alleles into non-autonomous gene drives whilst also expressing a putative antimalarial effector molecule. We sought to assess the efficacy of gene drive, the ability to express anti-Plasmodium molecules by linkage to genes active in infection-relevant tissues, and how these modifications would affect the expression of such host genes, as well as overall mosquito fitness. In order to establish this experimental proof-of-principle for IGD, we chose Scorpine as a prototypical effector molecule. This known anti-microbial peptide (AMP) had previously been shown to have a transmission blocking effect on *P. falciparum* in the context of paratransgenesis with several different microorganisms, and recently in transgenic mosquitoes in combination with other effectors (*Fang et al., 2011*; *Bongio and Lampe, 2015*; *Wang et al., 2012*; *Shane et al., 2018*; *Dong et al., 2020b*). For the same reason we focused here on female mid-gut specific genes since they allow targeting the *Plasmodium falciparum* parasite during the ookinete to oocyst transition stage (*Sinden, 2004*).

## Results

### Direct modification of three *A. gambiae* midgut loci

We chose CP (AGAP009593) and Aper1 (AGAP006795) for experimental validation of the IGD strategy because their promoters had been used previously for conventional transgene overexpression (*Ito et al., 2002*; *Abraham et al., 2005*). CP is expressed in the guts of pupae and sugar-fed adults and becomes 10-fold upregulated by blood-feeding as well as by feeding on protein-free meals, probably triggered by the gut distention (*Edwards et al., 1997*). Its mRNA levels peak 3 hr post blood-feeding (pbf), with spatial expression limited to the posterior midgut, and return to normal levels 24 hr pbf. Aper1 is only detected in the adult gut, but not in larvae, pupae, or adult carcass, and its mRNA expression profile is independent of age and blood-feeding (*Shen and Jacobs-Lorena, 1998*). The protein is stored in secretory vesicles and released into the midgut lumen soon after blood meal, where it is cross-linked into the peritrophic matrix via its two tandem domains that bind chitin (*Devenport et al., 2004*). The peritrophic matrix retracts from the midgut wall 48 hr pbf and is excreted after 72 hr pbf (*Dinglasan et al., 2009*). Using the VectorBase expression explorer (*Maccallum et al., 2011*) we identified further genes with significant expression in the female midgut and upregulation after the blood meal. Amongst them we chose alkaline phosphatase 2 (AP2, AGAP006400), which has a secretion signal and a GPI-anchor and is detected in the detergent resistant membrane (DRM) proteome of non-blood-fed adult *Anopheles gambiae* midguts (*Parish et al., 2011*).

We next identified guide RNA (gRNA) target sites around the start and stop codons of these genes for the insertion of the construct sequences (*Figure 1A*). The gRNAs were chosen based on a compromise between proximity of the cut site to the start or stop codons, activity and off-target scores, as well as the lack of common target site polymorphisms in public population genetic data sets of the mosquito (Table S1). We generated transformation constructs consisting of homology arms for these three loci designed to facilitate integration of the Scorpine coding sequence within each gene. In the case of CP and AP2, the Scorpine coding sequence was inserted at the start codon and linked to the coding sequence of the host gene via the 2A autocleavage peptide (*de Felipe and Ryan, 2004*). In contrast, we anchored the effector to the peritrophic matrix via a C-terminal fusion to Aper1. In the AP2 strain, Scorpine is expressed as a GFP-fusion with the aim to enhance stability of the protein (*Bokman and Ward, 1981*; *Janczak et al., 2015*). In each case, the effector coding sequences harboured an artificial intron encoding the gRNA (*Figure 1A*). We have recently characterized, in S12 cells and transgenic *Drosophila* strains, a number of artificial introns optimized equally for splicing and gRNA expression using an intronic RNA polymerase III promoter (A. Nash, unpublished) and we applied these designs here. Each artificial intron harbours a fluorescent marker driven by the 3xP3 promoter, flanked by loxP-sites, and the corresponding guide RNA under the control of the ubiquitous and constitutive *Anopheles gambiae* U6 promoter (*Figure 1A*).

For transgenesis, we co-injected a helper plasmid carrying the Cas9 coding sequence under the control of the vasa promoter (*Hammond et al., 2016*) and the results of these experiments are summarized in *Figure 1B and C*. We established the three transgenic strains Sco$^{GFP}$-CP, ScoG$^{CFP}$-AP2, and Aper1-Sco$^{GFP}$, in the latter case from a single G1 founder (*Figure 1B*). Integration of the vector backbone distinguished via the additional DsRed marker was only observed for the AP2 locus in several G1 individuals that were discarded (*Figure 1C*). Sequencing of all individuals within a G1 founder-cage for each strain confirmed that insertion into the genomic locus was precise at all three loci, and no aberrant integration events could be detected.

### Establishing minimal genetic modifications

We predicted that efficient splicing of the artificial introns would occur only in the absence of the 3xP3-GFP or CFP fluorescent transformation marker genes, as they contain the bidirectional SV40 terminator. We thus expected that the integrations we had established would interfere with the function of the mosquito host genes. However, following sib-mating we found that for all three modified loci, homozygous individuals were viable and fertile and showed no striking fitness defects during rearing and maintenance. In order to establish minimal genetic modifications, that is, to remove the fluorescent transformation markers flanked by loxP sites, we next crossed all three transgenic strains to a vasa-Cre strain (*Volohonsky et al., 2015*). We hereby established the markerless strains

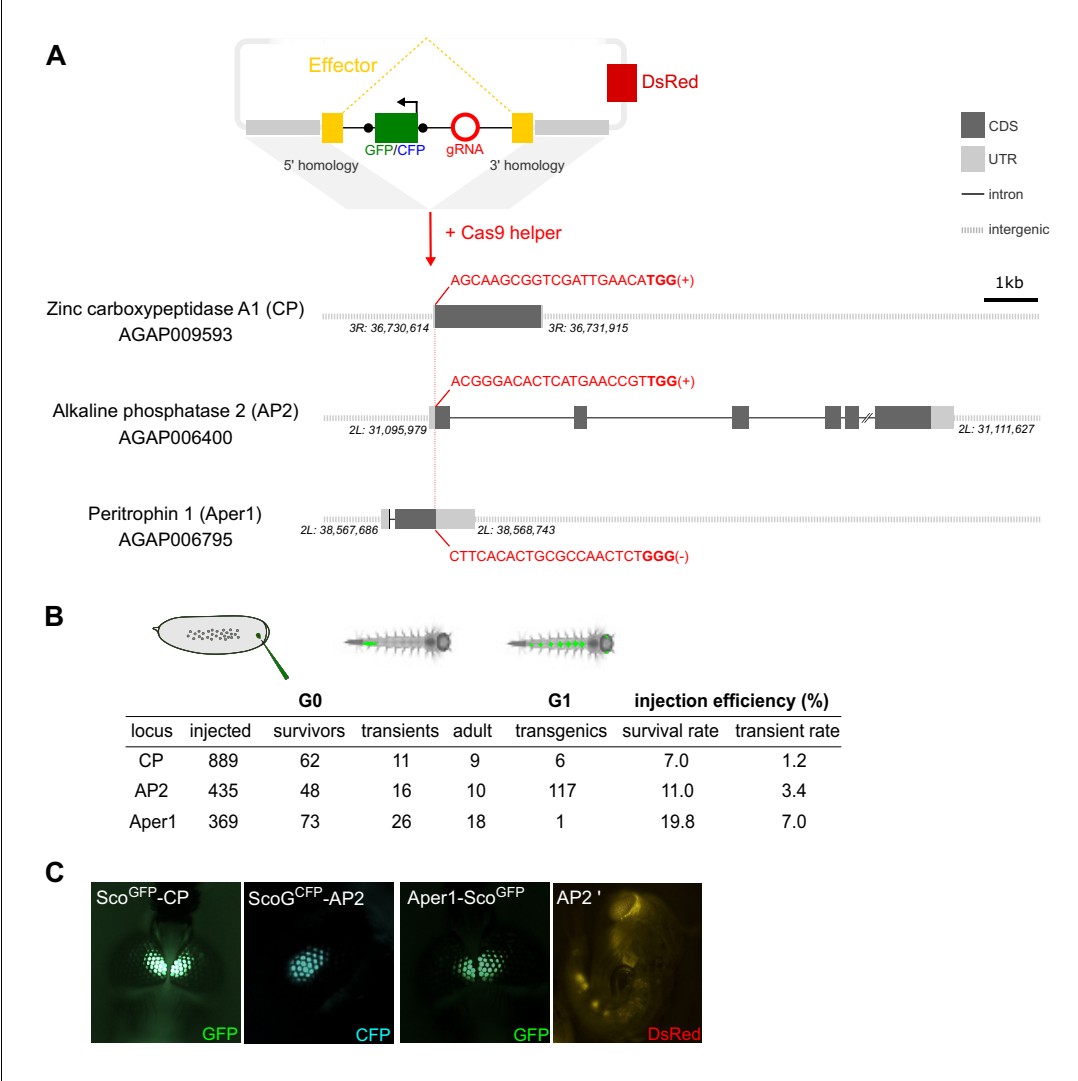

**Figure 1.** Homology-directed modification of *Anopheles gambiae* midgut loci by CRISPR/Cas9.  (**A**) Schematic showing the insertion of the effector construct at the carboxypeptidase (CP), alkaline phosphatase 2 (AP2), and peritrophin 1 (Aper 1) loci. The donor plasmid supplies the effector coding sequence (yellow), which accommodates an artificial intron. The intron harbours a fluorescent marker (either GFP or CFP, green) under the control of the 3xP3 promoter flanked by loxP sites (black dots) and a U6 driven guide-RNA module (red), required for both transgenesis and subsequently for gene drive. The plasmid features regions of homology that drive N- or C-terminal insertion at the start (CP, AP2) or stop codon (Aper1) as well as a 3xP3::DsRed plasmid-backbone marker. The gRNA target sequence (red) including the PAM motif (bold) and the target strand are indicated. (**B**) Summary of embryo microinjections and the identification of transgenic individuals by fluorescent screening. (**C**) Adult transgenic mosquitos with fluorescent expression of GFP or CFP in the eyes under the control of the 3xP3 promoter as well as a pupa showing DsRed fluorescence, indicating plasmid-backbone integration (AP2').

Sco-CP, ScoG-AP2, and Aper1-Sco (*Figure 2A*), all of which were also found to be homozygous viable and fertile. Establishing and tracking markerless modifications in *A. gambiae* are challenging and have not been attempted previously. Specifically, we achieved to establish pure-breeding markerless strains either by genotyping of pupal cases of individuals lacking visible fluorescent markers in the case of Sco-CP (*Figure 2B*) or in the case of Aper1-Sco by crossing the transhemizygotes to a Cas9 expressing strain to induce homing and preferential inheritance of the markerless allele (*Figure 2—figure supplement 1A*). For generating ScoG-AP2, we used the homozygous ScoG<sup>CFP</sup>-AP2 as a dominant balancing marker for tracking inheritance of the unmarked allele (*Figure 2—figure supplement 1B*). PCR on genomic DNA (*Figure 2C*) and subsequent sequencing confirmed precise removal of the marker module and also suggested successful homing of the construct.

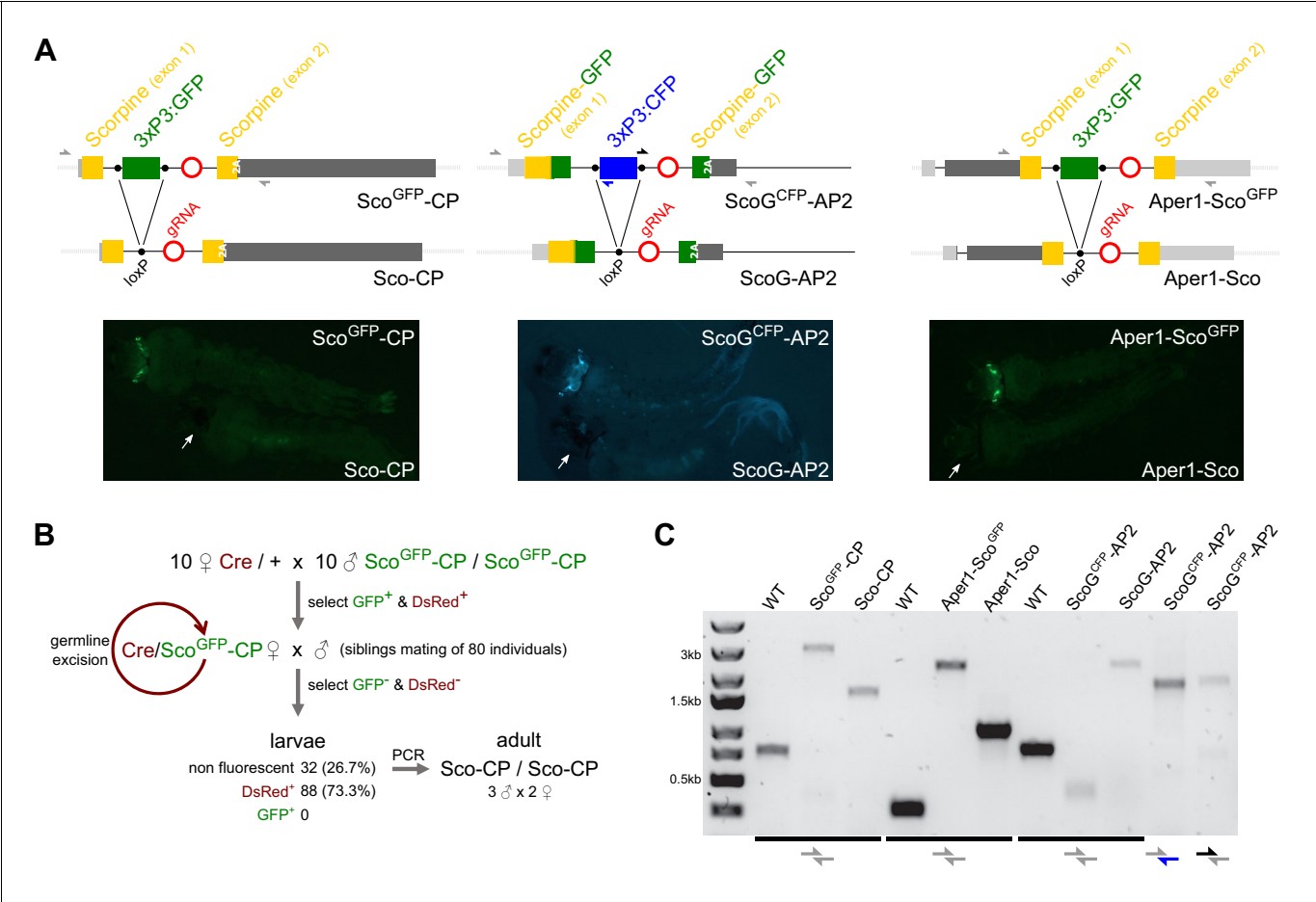

**Figure 2.** Generation of minimal genetic modifications. (**A**) Schematic showing the inserted transgene constructs within the exon structure of the CP, AP2, and Aper1 loci prior and following the excision of the marker gene by Cre recombinase (top) and the observed changes in green or cyan fluorescence in L3 larvae (bottom). Half arrows indicate primers for the PCRs shown in panel **C** and white arrows indicate the eyes in the markerless individuals. 2A indicates the F2A self-cleaving peptide signal. (**B**) Crossing scheme used for the establishment of markerless strain Sco-CP by crossing to a Cre recombinase expressing strain. Non-fluorescent adults were allowed to hatch individually, and their pupal cases were used for genotyping. (**C**) PCR genotyping of genomic DNA of homozygous individuals of all strains with primer-pairs (shown in **A**) spanning the three loci. The entire locus could not be amplified in strain ScoG^CFP-AP2 that contains both GFP and CFP, and hence separate 5' and 3' fragments were analysed by PCR.

The online version of this article includes the following figure supplement(s) for figure 2:

**Figure supplement 1.** Crossing schemes used for the establishment of markerless strains.

## Expression analysis of integrated effector sequences and their effect on the host genes

For a comparative expression study, we extracted RNA from female midguts 3 hr after blood-feeding of the three strains Sco^GFP-CP, ScoG^CFP-AP2, and Aper1-Sco^GFP and of the corresponding markerless strains Sco-CP, ScoG-AP2, and Aper1-Sco which carry minimal genetic modifications (*Figure 3A*). RT-PCRs over the splice junctions and subsequent sequencing confirmed that for all three markerless strains precise splicing occurs and that the removal of the artificial intron restores the open reading frame required for expression of both the effector protein and the host gene product (*Figure 3B*). However, we found evidence of full-length transcripts and successful splicing (possibly at reduced levels) even in the strains retaining the marker cassette which includes the bidirectional SV40 terminator (*Figure 3—figure supplement 1A*). This suggests that the modified host genes could be expressing the endogenous proteins to some degree even in these transgenic strains.

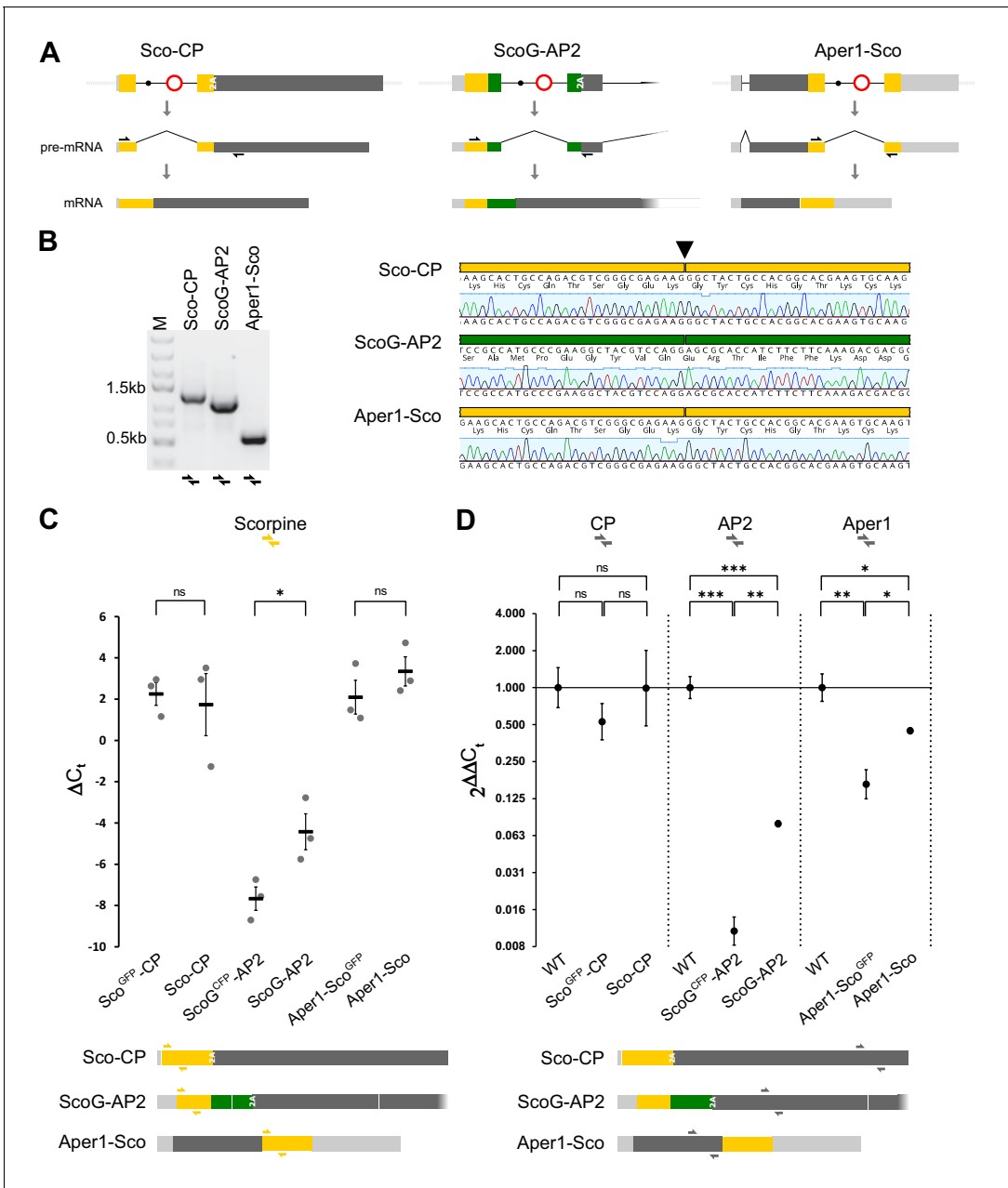

**Figure 3.** Splicing of the artificial intron and expression analysis. (**A**) Schematic showing mRNA expression of the modified host genes of the three markerless strains assuming correct splicing of the artificial intron. Black arrows indicate RT-PCR primers used in **B**. (**B**) RT-PCR (left) and sequencing of the amplicons (right) indicate precise splicing of the three artificial introns. Midguts of homozygous strains Sco-CP, ScoG-AP2, and Aper1-Sco were dissected 3 hr after blood feeding and RNA was extracted for RT-PCR. (**C**) Relative mRNA expression of Scorpine in the transgenic strains with and without the marker cassette. (**D**) Expression of the host genes CP, AP2, and Aper1 in transgenic strains with and without the marker cassette relative to the wild type. RNA was extracted from 10 to 15 midguts 3 hr after the blood meal in the case of CP and Aper1, and non-blood-fed for AP2. qPCR with the primer pairs indicated as coloured half arrows in the schematic below was conducted on cDNA and expression was normalized to the S7 house-keeping reference gene. Data derive from three biological replicates with three technical replicates each. p-values were calculated on $\Delta C_t$ values using the unpaired Student's t-test. *$p \leq 0.05$, **$p \leq 0.01$, ***$p \leq 0.001$, ****$p \leq 0.0001$, and ns: not significant.

The online version of this article includes the following figure supplement(s) for figure 3:

**Figure supplement 1.** Scorpine expression levels and mass spectrometry of co-opted host genes.

We next analysed mRNA expression by qPCR with a primer pair targeting exon 1 of Scorpine. In this preliminary experiment we compared non-blood-fed guts with guts dissected 3 hr after the blood meal for the three markerless lines (*Figure 3—figure supplement 1B*). In the case of ScoG-AP2, the qPCR signal of Scorpine was near the detection limit in blood-fed guts and low in non-blood-fed guts. For Sco-CP, an upregulation after the blood meal was observed in accordance with published data for the CP promoter (*Edwards et al., 1997*). Aper1-Sco showed the highest expression levels under both conditions. We thus continued our analysis with the 3 hr pbf time point for CP and Aper1 strains and with non-blood-fed guts for the AP2 strains. Using the Scorpine primers, no significant difference in the level of mRNA expression was found when comparing the Sco$^{GFP}$-CP and Aper1-Sco$^{GFP}$ with their respective markerless lines Sco-CP and Aper1-Sco (*Figure 3C*). A significant increase in expression was observed for strain ScoG-AP2 following removal of the fluorescent marker-module although overall expression levels were substantially lower when compared to the other two strains (*Figure 3C*). We further evaluated mRNA expression of the modified and unmodified host genes using primer pairs targeting the corresponding coding sequences of CP, AP2, and Aper1 (*Figure 3D*). Our analysis showed that host gene expression in strains Sco$^{GFP}$-CP and Sco-CP is not significantly different when compared to the wild type. The expression of AP2 and Aper1 is significantly reduced by the presence of the transgene including the marker cassette. Removal of the marker-module significantly restored host gene expression to approximately half of the level observed in the wild type in strain Aper1-Sco but remained below 10% for strain ScoG-AP2 (*Figure 3D*).

We also performed mass-spectrometry (LC-MS/MS) on non-blood-fed guts to confirm that protein expression of the co-opted host gene is unaffected by effector integration or splicing of the artificial intron. We detected high-confidence peptides for both genes in the Sco-CP and Aper1-Sco strains (*Figure 3—figure supplement 1B*). For Aper1, the peptides map to two out of the three regions from published data, and for Sco-CP 4 out of the 14 published peptides were identified (*Chaerkady et al., 2011*).

## Mosquito fitness analysis

We assessed the effects of these modifications on fecundity and larval hatching of all six transgenic strains and compared them to individuals of the wild-type (G3) colony and additionally to the vasa-Cre strain (*Volohonsky et al., 2015*), which derives from the *Anopheles gambiae* KIL background (*Meredith et al., 2011*) and was used to generate the markerless strains (*Figure 4*). While females of the strains including the marker-module showed no statistically significant difference compared to G3, the markerless strains Sco-CP, ScoG-AP2, and Aper1-Sco laid a significantly reduced number of eggs. The comparison suggests that here the modification of the host genes is not likely to be the main driver of the observed fitness effects. Since the removal of the fluorescent module by itself is unlikely to decrease fitness (or could even marginally improve it), inbreeding effects or the contribution of the KIL background (introduced via crossing to the Cre line) could partially explain the observed reduction in fecundity of the markerless strains. Indeed, the Cre strain also showed a significant decrease when compared to G3. ScoG$^{CFP}$-AP2 and Aper1-Sco strains additionally showed a statistically significant reduction in the hatching rate compared to the G3 wild type (*Figure 4C and D*). Since all three modified host genes are female and midgut specific, and because fitness effects would be expected to be sex-specific, we also measured pupal sex ratio of the markerless strains but found no significant deviations from an expected 1:1 sex ratio for any of the transgenics (*Figure 4—figure supplement 1*). Nevertheless, the presence of negative fitness effects of the introduced traits under these or other conditions could not be ruled out by our experiments.

## *P. falciparum* transmission blocking assays

We assessed the effect of these modifications on mosquito infection with the *P. falciparum* NF54 strain by feeding transgenic female mosquitoes on in vitro cultured gametocytes using a standard membrane feeding assay (SMFA) (*Habtewold et al., 2019*). Both the alteration and/or reduction of host gene expression as well as the expression of Scorpine could independently or jointly change the transmission characteristics of these strains. The number of oocysts in the midgut of wild-type and the homozygous transgenic strains was quantified 7 days pbf (*Figure 5*). We did not observe a significant effect on oocyst intensity or prevalence in strains ScoG$^{CFP}$-AP2 and Aper1-Sco$^{GFP}$

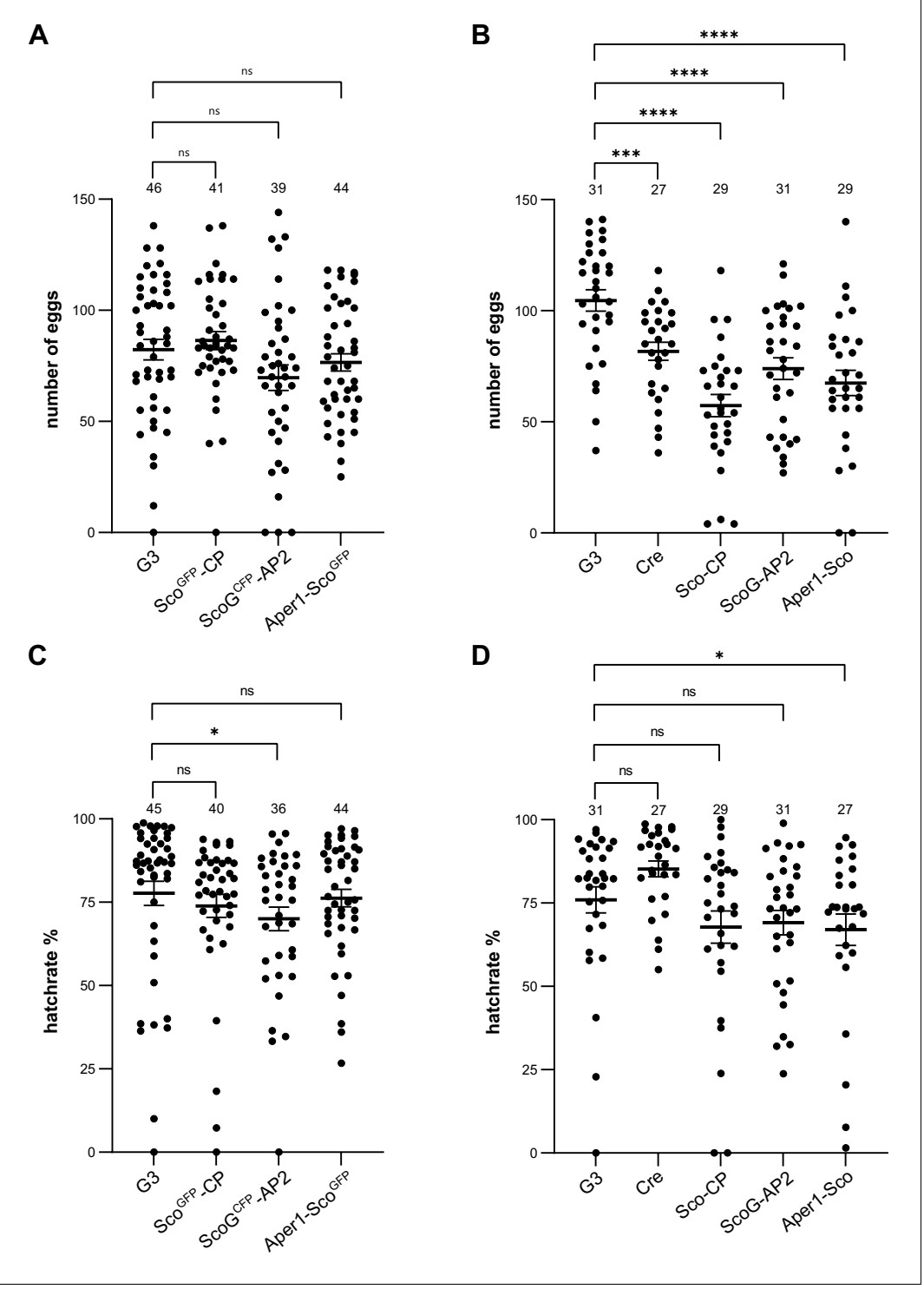

**Figure 4.** Fecundity and larval hatch rates of the transgenic strains. Fecundity of single females of the homozygous strains Sco$^{GFP}$-CP, ScoG$^{CFP}$-AP2, and Aper1-Sco$^{GFP}$(A) and the markerless strains Sco-CP, ScoG-AP2, and Aper1-Sco (B) compared to the G3 wild-type and the vasa-Cre strain (KIL background) used to remove the marker module. All data sets have a Gaussian distribution according to the Shapiro–Wilk test, except of Sco$^{GFP}$-CP. p-values were calculated using the unpaired two-tailed Student's t-test. Larval hatch rates of the transgenic strains with (C) and without (D) the marker, as well as the control strains, from the eggs above. None of the data sets showed a Gaussian distribution according to Shapiro–Wilk, and the p-values were calculated using the Kolmogorov–Smirnov test. Data from three biological replicates were pooled and the total number analysed is

*Figure 4 continued on next page*

indicated on top. Mean with standard error (SEM) were plotted. *p≤0.05, **p≤0.01, ***p≤0.001, ****p≤0.0001, and ns: not significant.

The online version of this article includes the following figure supplement(s) for figure 4:

**Figure supplement 1.** Pupal sex ratio.

(*Figure 5A*). This indicated that the observed reduction in host gene expression that occurs in both these strains had no substantial effect on transmission. Interestingly, strain Sco^GFP-CP showed a significant reduction in infection prevalence (IP; *Figure 5A*), whereas strain Sco-CP, on the contrary,

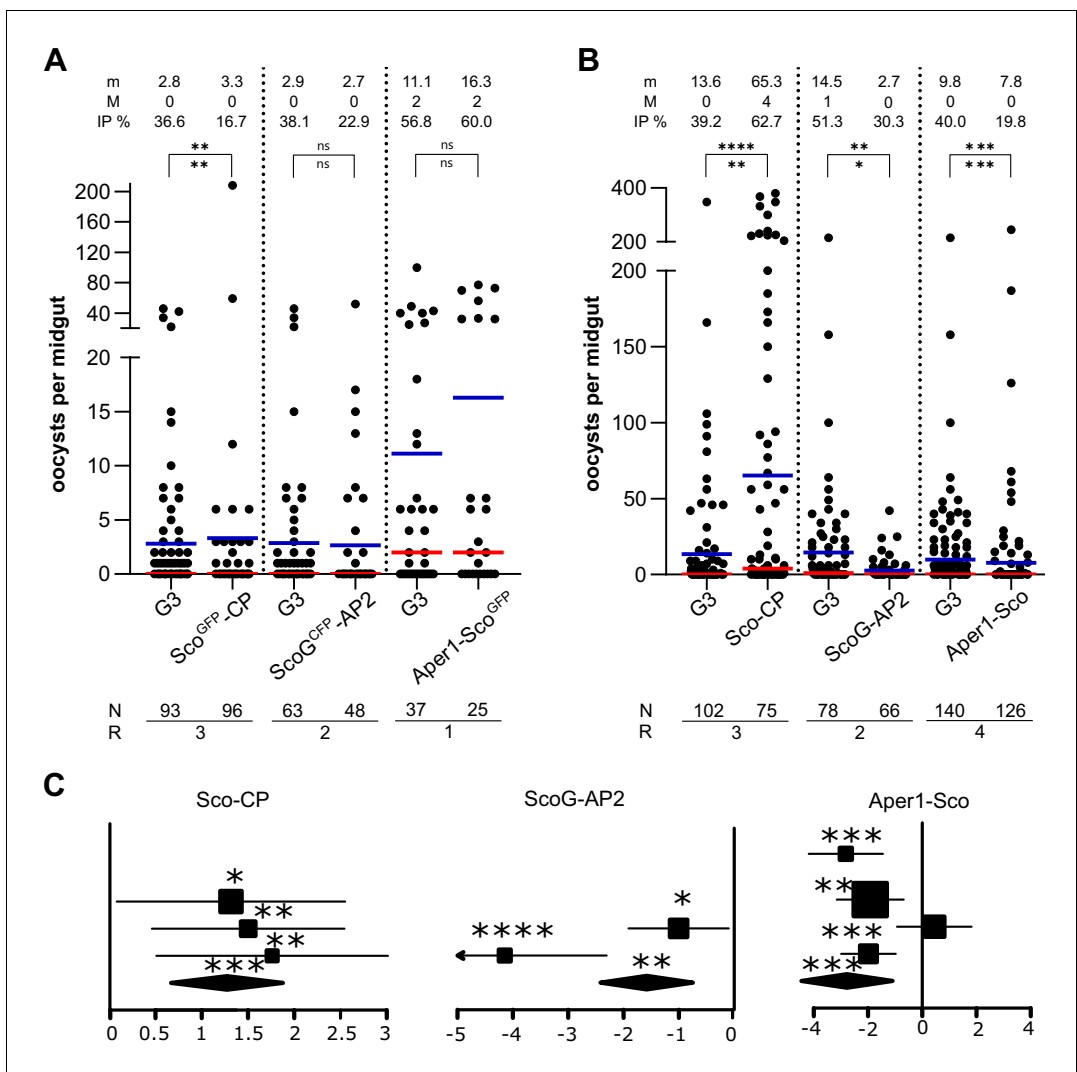

**Figure 5.** Transmission blocking assay. Standard membrane feeding assay with *P. falciparum* using the homozygous transgenic strains with the marker-module (A) and the corresponding markerless strains (B). Infection intensity is measured by the number of oocysts in each gut and the mean (m, blue bar) and median (M, red bar) are shown on top, as well as the infection prevalence (IP). The statistical significance of the infection intensity (stars above the bar) and IP (below) were calculated with the Mann–Whitney test and the chi-squared test, respectively. N is the number of mosquitoes analysed and R is the number of replicates performed. (C) Analyses of data plotted in (B) via a generalized linear mixed model (GLMM). The variation in the fixed effect estimates for each replicate (squares) and all replicates (diamonds) are shown as forest plots (95% confidence interval, glmmADMB). The square size is proportional to the sum of midguts analysed in each replicate. *p≤0.05, **p≤0.01, ***p≤0.001, ****p≤0.0001, and ns: not significant.

showed a significantly enhanced level of infection by both measures (*Figure 5B*). It had previously been shown that expression of two *A. gambiae* carboxypeptidase B genes is upregulated upon *P. falciparum* infection and that antibodies against one of them blocked parasite development (*Lavazec et al., 2007*). Similarly, antibodies against the *A. stephensi* carboxypeptidases A and B significantly reduced the oocyst number of *P. berghei* and *P. falciparum*, respectively (*VenkatRao et al., 2017*; *Raz et al., 2013*). A decrease in CP protein levels in strain Sco^GFP^-CP could possibly explain the observed effect. Strains ScoG-AP2 and Aper1-Sco both showed a significant reduction in *P. falciparum* oocysts (*Figure 5B*) in both IP and intensity.

## Analysis of non-autonomous gene drive induced by intronic gRNAs

Finally, we sought to determine whether the modified alleles of the three mosquito genes functioned as gene drives in the germline when provided with a source of Cas9 in trans (*Figure 6*). For these experiments we relied predominantly on the strains carrying the fluorescent marker in order to assay homing by scoring fluorescence in the progeny. We assessed homing efficiency of each gRNA under the control of the U6 promoter within the intron, by crossing the Sco^GFP^-CP, ScoG^CFP^-AP2 and Aper1-Sco^GFP^ strains to a vasa-Cas9 strain (E. Marois, unpublished). Transhemizygote individuals were subsequently crossed to the wild type and the rate of fluorescent progeny was recorded (*Figure 6A*). From this analysis, we excluded progeny that had also inherited Cas9 linked to the 3xP3-YFP marker (*Figure 6—figure supplement 1*). We calculated a homing rate of 95.95% and 98.43% for Sco^GFP^-CP and Aper1-Sco^GFP^, respectively, with slightly higher rates in females compared to males (Table S5). Strain ScoG^CFP^-AP2 reached a homing efficiency of 87.78%. Although this locus showed higher variability around the ATG insertion site in the Ag1000G data set (*The Anopheles gambiae 1000 Genomes Consortium, 2017*), no SNPs had been detected in 24 sequenced individuals from the G3 lab colony (*Bioproject Accession PRJNA397539, 2017*) suggesting that the lower homing rate was unlikely to be related to pre-existing resistant alleles.

We next sought to compare this result to the homing of a corresponding minimal genetic modification of the same locus. Homozygous strain ScoG-AP2 was crossed to the vasa-Cas9 strain and transhemizygotes were crossed to wild-type individuals. For this experiment, where homing could not be tracked via a fluorescent marker, the resulting progeny was assessed via PCR genotyping with primers amplifying the Scorpine coding sequence. We recorded near complete homing of 94.98% with 503 out of 516 individuals inheriting the ScoG-AP2 allele. Although this is an improvement compared to the homing rate of ScoG^CFP^-AP2, the difference was not statistically significant (t-test, p=0.1670).

To better understand the reason for the lower homing rate of ScoG^CFP^-AP2, we analysed non-fluorescent individuals by PCR genotyping (*Figure 6—figure supplement 2A*). For 73 out of 93 individuals analysed by primers spanning the entire locus, we detected a 2 kb product in addition to the wild-type amplicon. Sequencing of these PCR products suggested the construct was present but carried rearrangements that were likely the result of recombination between the GFP and CFP coding sequences (*Figure 6—figure supplement 2B*). Of the remaining individuals, half showed wild-type configuration at the guide RNA target site (*Figure 6—figure supplement 3B*). For Sco^GFP^-CP, 92% of the sequenced non-fluorescent individuals carried point-mutations or indels (*Figure 6—figure supplement 3B*). In contrast, all nine non-fluorescent individuals obtained from the experiments with Aper1-Sco^GFP^ retained the wild-type configuration (*Figure 6—figure supplement 3B*).

To obtain more quantitative data, we performed PCR amplicon sequencing using DNA from pooled offspring of the crosses of transhemizygotes with the wild type (*Figure 6B*). This experiment was designed to detect all possible modifications of the target sites, including those caused by zygotic/embryonic Cas9 activity either due to maternal or paternal deposition or, alternatively, due to somatic Cas9 activity in the 50% of the progeny that also inherit the Cas9 transgene. Embryonic Cas9 activity has previously been reported for the vasa promoter (*Hammond et al., 2016*). As the background we used the Cas9 strain and the G3 wild type and found that at all three loci the percentage of alleles called as variant was below 5% in these controls. In the progeny from crosses with female transhemizygotes the percentage of modified alleles was 73.7% for CP, 92.5% for AP2%, and 86.0% for Aper1, suggesting a significant effect of maternal deposition of Cas9 as previously described for the vasa promoter (*Papathanos et al., 2009*). In the offspring deriving from male transhemizygotes, we observed that the percentage of modified reads in the progeny was substantially lower than in the female cross but higher than expected if no Cas9 carryover or somatic Cas9

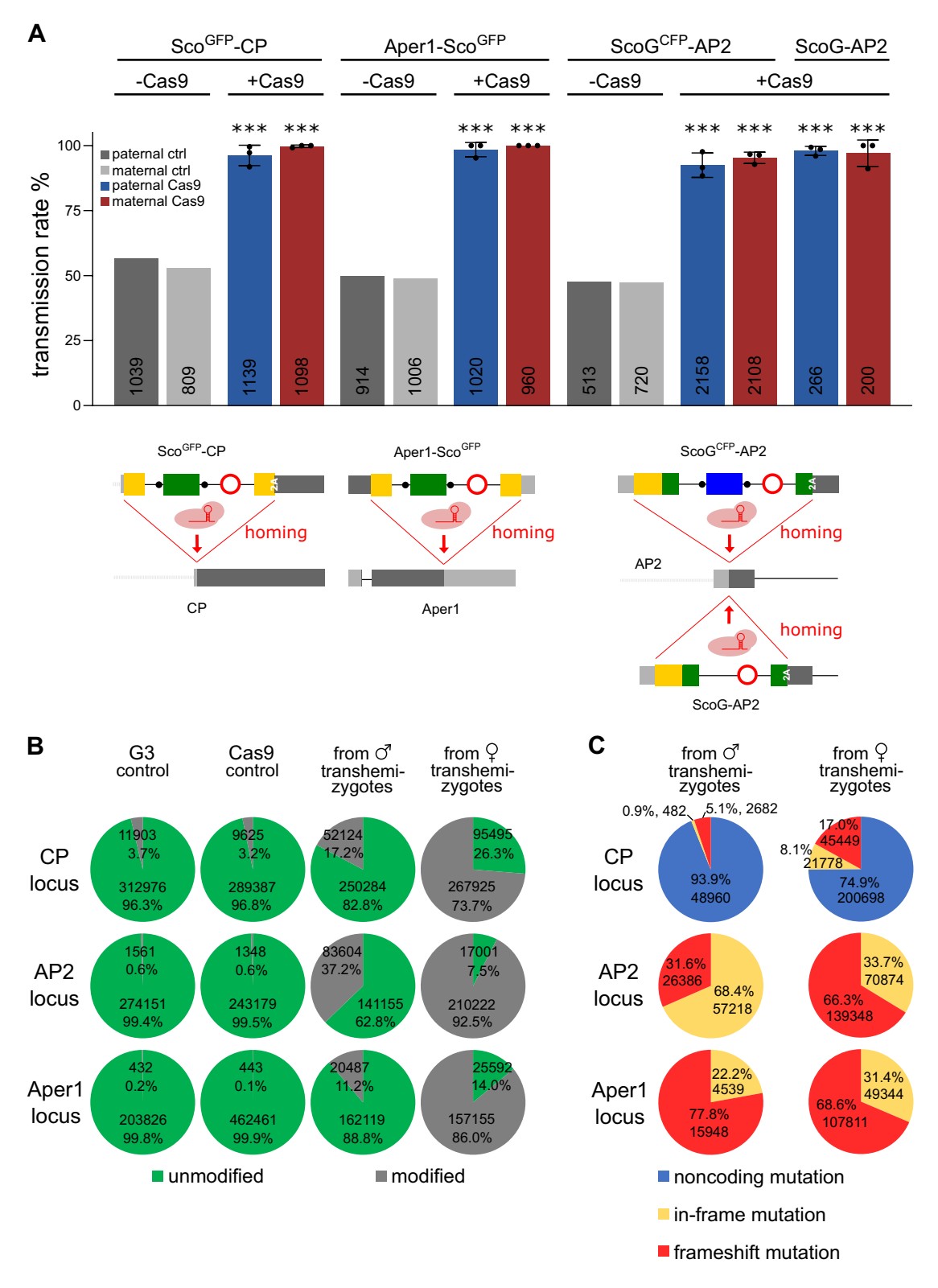

**Figure 6.** Assessment of non-autonomous gene drive of the modified host genes. (**A**) Homozygous individuals of strains Sco^GFP-CP, ScoG^CFP-AP2, and Aper1-Sco^GFP or the markerless strain ScoG-AP2 were crossed to the vasa-Cas9 strain to assess the homing potential induced by the intronic guide RNAs. As a control, hemizygous individuals lacking Cas9 were crossed to WT. In each case homing was measured by the rate of fluorescent larvae recorded in the progeny with the exception of the markerless strain ScoG-AP2 where it was assessed via PCR genotyping of the progeny. Mean and

*Figure 6 continued*

standard deviation from three biological replicates are plotted and the number n is indicated at the base of the column. All comparisons to the control crosses were significant (p<0.0001, chi-squared test). The data is found in *Supplementary file 1* - Supplementary Tables 4 and 5. (B) Amplicons from PCRs over the predicted cut site within the three loci performed on pooled progeny from transhemizygotes were subjected to next generation sequencing. G3 and Cas9 served as controls. Overall number of reads and the percentage of reads with modifications in the quantification window are shown. (C) Predicted classes of modifications represented within the set of modified alleles for each locus.

The online version of this article includes the following figure supplement(s) for figure 6:

**Figure supplement 1.** Crossing schemes for assessing the homing rate.
**Figure supplement 2.** Analysis of rearrangements in ScoG$^{CFP}$-AP2 following Cas9 cleavage.
**Figure supplement 3.** Sanger sequencing of non-fluorescent individuals.

activity is assumed to occur. We observed 17.2% modified reads for CP (against an expected maximum of 7.3% if activity is assumed to be restricted to the germline), 37.2% for AP2 (expected 7.6%), and 11.2% for Aper1 (expected 1.7%), suggesting embryonic Cas9 activity occurs in all three strains. For all three loci small indels at the cut site and towards the 5′ end of the gRNA were found to be responsible for the majority of modifications. When we predicted how these would affect the host gene coding potential, we observed distinct outcomes for the three loci (*Figure 6C*). At the CP locus, cleavage occurs upstream of the start codon and the majority of modifications were predicted to only affect the 5′ UTR. In contrast, cleavage of AP2 occurs downstream from the ATG, and hence the modifications observed are predicted to result in frameshifts or the loss of the start codon. Interestingly, the reciprocal crosses showed varying patterns of repair suggesting the different repair characteristics in the male and female germline or, alternatively, when repair follows after either germline or zygotic cleavage. At the Aper1 locus the effector was fused to the 3′ end of the coding sequence and the cut site is located 19 bases upstream of the STOP codon. The observed in-frame mutations (22.2% and 31.4% in paternal and maternal crosses) would thus affect the terminal amino acids only, whereas frameshifts were predicted to lead either to a premature STOP or to the addition of 25 (frame +2) or 26 (frame +3) C-terminal amino acids.

## Discussion

We modified three different genomic loci of *Anopheles gambiae* via precise CRISPR/Cas9 homology-directed integration and subsequently removed the fluorescent transformation marker cassettes using Cre recombinase mediated excision. A range of methods for the generation of markerless, genetically modified organisms have been developed in plants (*Tuteja et al., 2012*). The presence of such marker genes in genetically modified plants, and subsequently in food, feed, and the environment, are of concern and thus subject to special government regulations in many countries. With the view that gene drive strains that are currently being developed could eventually be deployed in the field and will have to undergo a stringent regulatory pathway and also that marker genes are known to induce negative fitness costs (*Catteruccia, 2003*; *Scolari et al., 2011*), the generation of minimal genetic modifications is paramount for moving gene drives towards application. Combinations of multiple gene drives and multiple fluorescent markers would not only compound such costs but also reduce the usefulness of such markers. Furthermore, the predictive power of marker genes in the field will be low as gene drives can decouple from any linked marker cassettes within one generation (*Oberhofer et al., 2018*). We show for the first time how markerless gene drive traits can be constructed in the malaria mosquito.

Unexpectedly, all strains including those carrying the entire fluorescent marker genes were found to be homozygous viable and fertile and showed no noticeable fitness cost during standard rearing conditions. We had assumed that removal of the marker would be necessary to avoid negative fitness consequences. RNA expression analysis indicated that host gene expression had indeed been reduced in strains ScoG$^{CFP}$-AP2 and Aper1-Sco$^{GFP}$ carrying the full insert including the fluorescent marker. The lack of a severe fitness phenotype could be explained by a certain redundancy in the set of digestive enzymes, proteases, phosphatases, and peritrophins expressed in the mosquito midgut, at least under standard rearing conditions, including a high-quality source of human blood. Alternatively, even low mRNA levels in conjunction with correct splicing of the entire intron including the marker cassette (we found evidence for this occurring in all three genes) could have led to

protein expression at a sufficient level, so as not to result in a measurable impact on fitness in our experiments. Following the removal of the marker gene, we found exact splicing of the artificial intron and a significantly restored expression level of the host genes carrying now only a minimal transgene insert. It remains to be seen whether the reduced levels of restored expression (in the case of AP2 and Aper1 loci) observed in this study would preclude the use of essential genes, for which expression levels are particularly critical, to host such IGDs. The intron we used derives from the *Drosophila melanogaster* homeobox transcription factor *fushi tarazu* (*ftz*), essential for early embryonic development, and we have separately found that a full gRNA transgene can be hosted within this intron (A. Nash, unpublished). Our data here show similarly that the *ftz* intron can accommodate the *A. gambiae* U6 promoter and the gRNAs for gene drive and remains functional in mosquitoes. The splicing was precise, in the context of Scorpine (AAG-intron-GG) or when inserted within Scorpine::GFP (AGG-intron-AG).

When we provided a source of Cas9 in the male or female germline, all gRNA transgenes were able to induce high rates of gene drive of the modified loci. In the case of the AP2 locus we found that the minimal, markerless locus ScoG-AP2 outperformed ScoG$^{CFP}$-AP2 although the difference was not statistically significant. We observed aberrant homing events in the latter strain, which are likely the result of recombination between the GFP and CFP coding sequences. This highlights that simple genetic modifications may avoid undesired interactions found in more complex constructs. The interaction of multiple gene drives (sharing homologous sequences such as marker genes, promoters, or terminators) could thus lead to unintended outcomes in the field. We analysed the formation of resistance alleles at these loci, including due to zygotic activity of vasa-driven Cas9 and predicted distinct outcomes for the three target loci. While the host genes we targeted appeared not to be essential (with the above-mentioned caveat that sufficient expression may occur in our strains), clear negative effects may result from their genuine disruption, for example, by a frameshift. Further studies for example in population cages could thus reveal to what degree the observed classes of mutations (*Figure 6C*) would be actively selected against, which could favour faithful transmission of the drive (*Nash et al., 2019*).

While our amplicon-sequencing experiments could not distinguish paternal carryover of Cas9 from somatic Cas9 activity (in male individuals inheriting both transgenes) they clearly showed an increased level of non-homologous repair in progeny of transhemizygous females indicating maternal deposition of Cas9. It should be noted that non-autonomous effectors as described herein could also be combined with male-specific gene drives, suppressive gene drives, or autonomous rescue drives and the rate of the formation and the dynamic of selection of functional resistance alleles via these routes would be quite different in each case.

The regulatory regions of the target genes CP and Aper1 had previously been used for mosquito transgenesis (*Ito et al., 2002*; *Abraham et al., 2005*). In contrast, AP2 chosen due to differential expression in the female gut has not been described before. In our experiments, overall expression from the AP2 locus was found to be lower when compared to the CP and Aper1 loci, suggesting that AP2 would be less suitable as a host gene or as a transgene promoter for the expression of effectors requiring high levels of protein. Strong promoters might not always be ideal however, as the dosage-response curves of other types of effectors (e.g. transcriptional modulators) may vary significantly. Together, our data on all three genes suggest, that co-opting the regulatory elements of endogenous loci directly without prior labour-intensive characterization is a viable approach. Since there is a dearth of promoters for many infection-relevant tissues (e.g. salivary glands, hemocytes, fatbody) we propose that this characterization step could be avoided, and effector molecules directly incorporated into mosquito gene loci and mobilized by non-autonomous drive. Existing RNA expression data sets or enhancer-trap screens could provide a starting point for the identification of suitable genes (*Reid et al., 2018*).

Our analysis suggested however that modifications of endogenous genes could also have unforeseen consequences. In the case of the zinc carboxypeptidase A1 (CP) we found that modifications of the host gene are possibly directly altering infection outcomes. The reduced infection level observed for strain Sco$^{GFP}$-CP which may express reduced levels of CP protein is consistent with reports on other carboxypeptidase genes (*Lavazec et al., 2007*; *VenkatRao et al., 2017*; *Raz et al., 2013*). In contrast, strain Sco-CP showed a positive effect on infection levels which would suggest that the CP locus may not be a suitable host gene for gene drives. We observed a reduction in *P. falciparum* transmission in strains ScoG-AP2 and Aper1-Sco. Scorpine was chosen for our study as a prototypical

effector molecule as it had previously shown to reduce Plasmodium survival (*Fang et al., 2011*; *Bongio and Lampe, 2015*; *Wang et al., 2012*; *Shane et al., 2018*) when expressed via different routes in the mosquito midgut. We do not know how the midgut environment affects the stability of the lysine-rich Scorpine and detection of even the host proteins in these samples proved challenging. Modifications of the host genes AP2 and Aper1 did not appear to alter infection levels on their own. Overall, the moderate effects on transmission we observed in different directions in our pilot experiments call for further transmission studies with these strains in order to determine whether Scorpine has realistic potential as an antimalarial effector and to lend further support to the suitability of the three mosquito genes to host effector molecules. Both genetic polymorphisms linked to the modified loci in these inbred strains and the effect of the genetic backgrounds used could influence parasite infections and our transmission experiments were not designed to distinguish these effects in full. In addition, determining the effect of these modifications on later stages of parasite development using molecular assays is also necessary to obtain a full picture. Strains Sco$^{GFP}$-CP, ScoG$^{CFP}$-AP2, and Aper1-Sco$^{GFP}$ could also be used to isolate defined NHEJ and knockout alleles of the three host genes and characterize their effect on transmission and fitness. Such studies will likely be required to demonstrate the safety of this and related approaches utilizing host genes.

In summary, the strategy and the selected host genes we have presented here now allow for the insertion and direct comparisons of a range of different antimalarial effectors either by secretion into the gut or by anchoring these molecules in the peritrophic matrix. Such effectors traits would then by design, as we have shown here, be capable of efficient non-autonomous gene drive. The most significant knowledge gap, when it comes to transmission blocking modifications of transgenic mosquitoes, is to what degree any effects observed with lab strains of *P. falciparum* would be reproducible using circulating parasites isolated from patient blood or would hold up under actual field conditions which could wildly differ from standard laboratory experiments. The integral effector designs we have described could be valuable tools to attempt to answer this question. Since they are incapable of autonomous gene drive, they can be established and maintained, unlike suppression drives where this is impossible, as true-breeding strains. Importantly, the regulatory burden for importing or generating such non-autonomous effector strains would be expected to be much lower than strains capable of full gene drive and hence, simplify advanced stages of testing against polymorphic isolates of the *P. falciparum* parasite in the endemic setting without the need for strict containment or geographical isolation. When combined with other strains capable of autonomous Cas9 gene drive, whether designed for replacement or suppression, these strains could then, without the need for any further genetic modification, be deployed to contribute to reducing malaria transmission by mosquito populations in the field.

# Materials and methods

## Plasmids and primers

The two intermediate plasmids pI-Scorpine-GFP and pI-Scorpine were generated first to allow insertion of different gRNAs. We synthesized the Scorpine coding sequences, optimized for codon usage in *Anopheles gambiae* and retained the endogenous secretion signal from Scorpine. The GFP containing the intron, based on the *Drosophila melanogaster fushi tarazu* (ftz) intron, was amplified from pQUAST 3xP3 CFP (A. Nash, unpublished) and the U6 promoter was exchanged for the *A. gambiae* U6 spliceosomal RNA (AGAP013695) promoter from p165 (*Hammond et al., 2016*). The EGFP marker module within pI-Scorpine has a 3xP3 promoter and a SV40 terminator and was amplified from p163 (*Hammond et al., 2016*). The intron within Scorpine was inserted between the nucleotides AAG and G to promote splicing (*Schwartz et al., 2008*). We used the Drosophila splice predictor (*Reese et al., 1997*) to remove cryptic splice sites. A furin cleavage site and the F2A peptide (*Galizi et al., 2014*) were generated via annealing long oligos (53-F2A-long-F and 54-F2A-long-R). Some of the initial fragments were pre-fused via overlap extension PCR in order to minimize the number of parts to be fused for the final Gibson assembly.

The guide-RNAs for the three target loci were generated via the annealing of oligonucleotides 39-CP-gRNA59-F and 40-CP-gRNA59-R, 57-AP2-gRNA53-F and 58-AP2-gRNA53-R, as well as 75-Per1-gRNA48-F and 76-Per1-gRNA48-R. They were then inserted into the intermediate plasmids through the BbsI restriction site via Golden gate cloning. Subsequently, each effector-cassette

hosting the corresponding gRNA was PCR-amplified from the resulting intermediate plasmids. For the donor plasmids pD-ScoG-AP2 and pD-Sco-CP, the cassette was amplified with primers 51-Scorpine-F and 6-F2A-R, and for pD-Aper1-Sco with primers 77-Arg-GSG-Arg-Scorp-F and 97-Scorpine-STOP-R.

The 5' and 3' homology arms of approximately 800 bp were amplified from *A. gambiae* G3 genomic DNA. All amino acid changes detected in protein sequences were confirmed via the variation data available in http://www.vectorbase.org, except the A23V amino acid change in the secretion signal of the CP CDS. Nevertheless, this variant was observed in the 24 individual G3 sequencing reads. The guide RNA target present in the Aper1 CDS 5' of the STOP codon was destroyed by re-coding five wobble bases in the Aper1 5' homology arm.

The donor plasmids pD-Sco-CP, pD-ScoG-AP2, and pD-Aper1-Sco (full plasmid sequences are provided in *Supplementary file 2*) were generated by assembling the homology arms, the cassette, and the backbone containing an additional 3xP3-DsRed marker module. All plasmids were propagated in Agilent Sure cells in order to avoid recombination. All gRNAs were evaluated in vitro with the Guide-it sgRNA In Vitro Transcription and Screening Kit (Takara), and were found to cleave efficiently. For primers and plasmids see Tables S2 and S3.

## Microinjections and establishment of transgenic strains

Plasmids were isolated with the ZymoPURE II Plasmid Maxiprep kit (Zymo Research) and 300 ng/µL of the donor plasmid and the p155-helper-plasmid (*Hammond et al., 2016*) were microinjected into 30–45 min old eggs of *A. gambiae* G3 as described previously (*Lobo et al., 2006*). Hatching larvae were screened for transient expression of the fluorescent marker and the adults crossed to wild type. The G1 offspring were screened for the presence of the GFP (Sco$^{GFP}$-CP, Aper1-Sco$^{GFP}$) or CFP (ScoG$^{CFP}$-AP2) markers and we selected against red fluorescence in the eyes, in order to exclude individuals with possible plasmid backbone integration events, which was only observed in the case of ScoG$^{CFP}$-AP2. G1 transgenic founder adults were backcrossed to WT and following egg laying, they were sacrificed and genomic DNA isolated. We performed PCR over the 5' and 3' insertion points using one primer binding within the construct and another primer binding the flanking genome sequence beyond the homology arms. Genomic DNA was obtained from single adults via crushing them in 100 µL of 5% w/v Chelex100-resin beads (BioRad Inc) in water and 4 µL of 600 U/mL Proteinase K. After incubation at 55°C and 750 rpm for 2 hr, the Proteinase was heat inactivated at 99°C for 10 min, the samples were centrifuged at maximum speed for 3 min and the supernatant was transferred to a new tube. All genotyping was performed with RedTaq Polymerase (VWR). Since for ScoG$^{CFP}$-AP2 we obtained more than 100 G1 transgenic founder individuals, 12 single crosses were prepared in cups and offspring pooled after the parents were genotyped by sequencing. The transgenic founders confirmed by sequencing were subsequently outcrossed to G3 WT for three generations.

## Establishment of pure breeding transgenic strains

Sco$^{GFP}$-CP and ScoG$^{CFP}$-AP2 were rendered homozygous by setting up sibling crosses over two generations and letting the adults hatch single in cups and genotyping their left-over pupal cases by PCR over the entire locus to distinguish WT, heterozygotes, and homozygotes. Sco$^{GFP}$-CP was genotyped with primers 99-CP-locus-F and 100-CP-locus-R, and ScoG$^{CFP}$-AP2 with primers 101-AP-locus-F and 102-AP-locus-R. In the case of Aper1-Sco$^{GFP}$, the transgene was crossed to the vasa-Cas9 strain and kept in this background for several generations to allow for homing to occur. The YFP marker is also visible in the green channel and was removed via screening for orange fluorescence (excitation 515–545 nm, emission 585–670 nm). Afterwards, crosses of four females to four males were prepared in cups and the parents were genotyped with 230-Per-short-F and 231-Per-short-R after egg laying. gDNA was isolated from pupae cases in 20 µL dilution buffer of the Phire Tissue Direct PCR Master Mix kit (Thermo Scientific).

## Removal of the marker module using Cre recombinase

Markerless strains were generated by removal of the fluorescent marker cassette through crossing to strain C2S (*Volohonsky et al., 2015*) expressing Cre recombinase driven by the vasa promoter and carrying a 3xP3-DsRed marker (herein referred to simply as Cre). The Cre transgene is on the

third chromosome and resides in a KIL background and contains an additional 3xP3-CFP marker (*Meredith et al., 2011*). In the case of CP, the crossing scheme was initiated with homozygous Sco<sup>GFP</sup>-CP individuals and offspring showing green fluorescence and red fluorescence for Cre were kept for setting up a siblings-cross (*Figure 2B*). CFP and GFP fluorescence were separated by employing a narrower CFP filter-set (ET436/20x, ET470/24m, T455LP, Chroma) and a YFP filter-set (ET480/20x, ET525/50m, T495LP, Chroma). The progeny from Sco<sup>GFP</sup>-CP/Cre were screened against the presence of both markers and 88 larvae still containing the Cre transgene were discarded. There were no more larvae with GFP present, and hence, the efficiency of the Cre-mediated excision was 100%. Thirty-two individuals were markerless and were genotyped by PCR on pupal cases for homozygosity. Seven individuals were homozygous, three hemizygous, one WT, and 18 PCRs had unclear results. A pure breeding strain was established immediately afterwards from three male and two female markerless founders.

In the case of Aper1, the crossing scheme was started with hemizygous Aper1-Sco<sup>GFP</sup> individuals and the offspring were screened for GFP and DsRed (*Figure 2—figure supplement 1A*). After crossing them to the Cas9 strain, the progeny was screened against CFP to remove the Cre transgene but were kept in this Cas9 background over several generations, in order to make them homozygous. After removal of the Cas9 transgene, 13 final founders were confirmed via genotyping and sequencing over the lox-out, as well as over the 5' and 3' insertion sites and no imprecise homing events were detected.

In the case of the AP2, we used homozygous ScoG<sup>CFP</sup>-AP2 individuals to initiate the cross to the Cre strain which also features the 3xP3-CFP marker (*Figure 2—figure supplement 1B*). Offspring exhibiting both blue and red fluorescence were crossed to the vasa-Cas9 strain and the progeny were screened for the presence of YFP and the absence of the CFP and DsRed markers, which would maintain the markerless transgene in a Cas9 background. One further generation was propagated without screening in order to let the cassette home. Subsequently, individuals were genotyped using pupal cases and homozygotes were crossed to homozygous ScoG<sup>CFP</sup>-AP2 individuals. The offspring were screened against the presence of the YFP to select against Cas9. Finally, ScoG-AP2/ScoG<sup>CFP</sup>-AP2 mosquitoes were crossed, and the progeny were screened against CFP. The majority of them was also genotyped after egg laying.

## RT-PCR and qPCR

WT and transgenic mosquitos were dissected 3 hr after blood-feeding and tissue from 10 to 15 guts homogenized with glass beads for 30 s at 6,800 rpm in a Precellys 24 homogenizer (Bertin) in Trizol. RNA was extracted with the Direct-zol RNA Mini-prep kit (Zymo Research) and converted into cDNA with the iScript gDNA Clear cDNA Synthesis Kit (Bio-Rad). RT-PCR was performed with RedTaq Polymerase (VWR) with primers 51-Scorpine-F and 117-CP-ctrl-R on Sco-CP (1267 bp), 160-Sco-probe-F and 6-F2A-R on ScoG-AP2 (1089 bp), and 105-Per-locus-F and 162-Sco-probe-2-R on Aper1-Sco (533 bp). The products were subsequently evaluated via gel-electrophoresis and sequencing (Genewiz). RNA was isolated from three biological replicates and qPCRs were performed in triplicates for each with the Fast SYBR Green Master Mix (Thermo Scientific) on an Applied Biosystems 7500 Fast Real-Time PCR System. Expression was normalized to the S7 reference gene and analysed using the ΔΔCt method. The following primer pairs were used for the qPCRs: 270-q-CP-F1 and 271-q-CP-R3, 450-qAP-F3 and 451-qAP-R3, 274-qPer1-F and 275-qPer1-R, as well as 213-q-Sco1-F1 and 523-qSco1-all-R (Table S2).

## Mass spectrometry

Following dissection, 25 non-blood-fed guts of strains Sco-CP or Aper1-Sco were put in 50 µL CERI reagent from the NE-PER Nuclear and Cytoplasmic Extraction kit (Thermo Fisher Scientific), homogenized with a motorized pestle and cytoplasmic proteins were extracted according to the manufacturer's protocol. The supernatant was filtered through a 100 kDa or 50 kDa Amicon Ultra-0.5 Centrifugal Filter Unit for Sco-CP and Aper1-Sco, respectively. Trypsin-digest and LC-MS/MS were performed at the Advanced Mass Spectrometry Facility at the University of Birmingham.

## Fitness assays

For fecundity and fertility assays, single blood-fed females were transferred to cups containing water, lined with filter paper. Females that failed to lay eggs or produce larvae were dissected and excluded from the analysis if no sperm was detected in their spermatheca. Eggs and L1 larvae from at least 10 technical replicates were counted and the data from three biological replicates were pooled. All biological replicates passed the Shapiro–Wilk normality test, except of Sco$^{GFP}$-CP. To determine the pupal sex ratio, between 100 and 140 L1 larvae per tray were reared to the pupal stage and sexed. Data from three biological replicates was pooled and analysed for deviations from the expected 1:1 sex ratio via the chi-squared test.

## *Plasmodium falciparum* standard membrane feeding assay

Infections of mosquitoes using the streamlined standard membrane feeding assay were performed as described previously (*Habtewold et al., 2019*). Briefly, mosquitoes were fed for 15 min at room temperature using an artificial membrane feeder with a volume of 300–500 µL of mature *Plasmodium falciparum* (NF54) gametocyte cultures (2–6% gametocytaemia). Afterwards, mosquitoes were maintained at 26˚C with 70–80% relative humidity. For 48 hr after the infective meal, mosquitoes were deprived of light and starved without fructose, in order to eliminate the unfed mosquitoes. Midguts were dissected after 7 days and oocysts were counted to calculate mean and median (including all zero values). Statistical significance was calculated using the non-parametric Mann–Whitney test for oocyst load (infection intensity) and the chi-squared test for oocyst presence (IP) with GraphPad Prism v7.0. The generalized linear mixed model (GLMM) was used to determine statistical significance of oocyst infection intensity for each independent biological replicate. GLMM analyses were performed in R (version 1.2.5019) using the Wald Z-test on a zero-inflated negative binomial regression (glmmADMB). The different strains were considered as covariates and the replicates as a random component. Fixed effect estimates are the regression coefficients.

## Assessment of the homing rate

Homozygous individuals of strains Sco$^{GFP}$-CP, ScoG$^{CFP}$-AP2, and Aper1-Sco$^{GFP}$ were crossed to the vasa-Cas9 strain carrying the 3xP3-YFP marker (chromosome 2) and the offspring were screened for the presence of orange fluorescence (*Figure 6—figure supplement 1*, Table S5). Subsequently, the transhemizygotes were sexed and crossed to G3 wild type. Offspring deriving from the ScoG$^{CFP}$-AP2 cross were screened for the presence of CFP, whereas progeny deriving from Sco$^{GFP}$-CP and Aper1-Sco$^{GFP}$ were selected based on the absence of orange fluorescence and the presence of the GFP marker.

As controls, hemizygous individuals of each strain were crossed to WT, and the rate of fluorescent larvae was calculated. For the calculation of the homing rate e, we used the individuals negative for the effector ($E_{neg}$), the number n, and the Mendelian distribution of 50% as a baseline, as follows:

e = (n*0.5 – $E_{neg}$) / (n*0.5) *100.

In order to evaluate the homing rate of the markerless AP2-line, transhemizygote offspring were screened by PCR for the presence of the construct. gDNA was isolated with Chelex beads (BioRad) or the dilution buffer of the Phire Tissue Direct PCR Kit (Thermo Scientific). For the first biological replicate, PCRs were performed with primers 101-AP-locus-F and 102-AP-locus-R, and in case of doubt another PCR with primers 160-Sco-probe-F and 161-Sco-probe-R was performed. For the second and third biological replicates, a multiplex PCR with primers 101-AP-locus-F, 102-AP-locus-R, and 273-qSco1-R2 was conducted.

Non-fluorescent individuals were subjected to PCR with primers 99-CP-locus-F and 100-CP-locus-R, 101-AP-locus-F and 102-AP-locus-R, and 230-Per1-short-F and 231-Per1-short-R, respectively. CRISP-ID (*Dehairs et al., 2016*) was used for the deconvolution of the Sanger-sequencing chromatograms.

## Amplicon sequencing

Genomic DNA was extracted from all offspring of the respective homing crosses at larval stage L2 with the Monarch Genomic DNA Purification Kit (NEB). The vasa-Cas9 strain and G3 wild type were included as controls. PCRs over the insertion sides using Q5 High-Fidelity DNA Polymerase (NEB) were performed with primers 99-CP-locus-F and 524-CP-ampli-R (443 bp amplicon), A3-AP-

PrimerC-F and 221-q-AP-R (488 bp amplicon), as well as 230-Per-short-F and 231-Per-short-R (303 bp amplicon). An extension time of 5 s was chosen to exclude transgene amplification. Annealing temperature and cycle number were set to 66°C and 25 for CP and Per, and 60°C and 29 for AP, respectively. Amplicons were purified with QIAquick PCR Purification Kit (Qiagen) and submitted to Amplicon-EZ NGS (Genewiz). Raw NGS data (SRA accession PRJNA701314) were analysed with CRISPResso2 (*Clement et al., 2019*) and the minimum average read quality score (phred33) was set to 30.

## Acknowledgements

We thank Alexander Nash, Roberto Galizi, and Andrew Hammond for template-constructs, George Avraam and Olivia Bates for help with line maintenance, Claudia Wyer for technical assistance, Louise Marston and Carla Siniscalchi for injection training, Eric Marois for sharing the vasa-Cre-line and vasa-Cas9-line, and Jinglei Yu from the Advanced Mass Spectrometry Facility at the University of Birmingham. This work was funded by the Bill and Melinda Gates Foundation grant OPP1158151 to NW and GKC.

## Additional information

### Funding

| Funder | Grant reference number | Author |
| --- | --- | --- |
| Bill and Melinda Gates Foundation | OPP1158151 | George K Christophides<br>Nikolai Windbichler |

The funders had no role in study design, data collection and interpretation, or the decision to submit the work for publication.

### Author contributions

Astrid Hoermann, Conceptualization, Data curation, Formal analysis, Investigation, Visualization, Writing - original draft, Writing - review and editing; Sofia Tapanelli, Data curation, Formal analysis; Paolo Capriotti, Resources, Data curation; Giuseppe Del Corsano, Resources, Data curation, Investigation; Ellen KG Masters, Data curation; Tibebu Habtewold, Conceptualization, Data curation, Supervision, Funding acquisition, Project administration, Writing - review and editing; George K Christophides, Conceptualization, Supervision, Funding acquisition, Visualization, Writing - original draft, Project administration, Writing - review and editing; Nikolai Windbichler, Conceptualization, Resources, Supervision, Funding acquisition, Investigation, Visualization, Writing - original draft, Project administration, Writing - review and editing

### Author ORCIDs

Astrid Hoermann (ID) https://orcid.org/0000-0002-1346-5945
Nikolai Windbichler (ID) https://orcid.org/0000-0001-9896-1165

### Decision letter and Author response

Decision letter https://doi.org/10.7554/eLife.58791.sa1
Author response https://doi.org/10.7554/eLife.58791.sa2

## Additional files

### Supplementary files

• Supplementary file 1. Supplementary Tables S1–S7. Table S1. Guide RNA and target site characteristics. The start codon of the target gene is indicated within the gRNA sequence (bold) and the protospacer adjacent motif (PAM) separated via a hyphen. All predicted off-target cleavage sites were found to be located in non-coding (NC) regions and the number of mismatches (MM) is indicated. The number of SNPs within the 24 individuals of the G3 strain and within the Ag1000G is indicated. The SNPs observed for CP in the Ag1000G did not pass the quality control. Table S2. Primers

used in this study. Table S3. Plasmids used in this study. Table S4. Transmission rate of control-crosses without Cas9. Table S5. Transmission rates and homing rates. $E_{pos}$ and $E_{neg}$ refer to individuals with or without the effector construct, respectively. The homing rate e was calculated as follows: e = (n*0.5–$E_{neg}$)/(n*0.5)*100. Table S6. Modified sequences identified in the Amplicon sequencing. Table S7. Raw data of the transmission blocking assay.

- Supplementary file 2. GenBank-DNA-files of donor plasmids pD-Sco-CP, pD-ScoG-AP2, and pD-Aper1-Sco.

- Transparent reporting form

### Data availability

All data generated or analysed during this study are included in the manuscript and supporting files. Amplicon-sequencing raw data have been deposited in SRA under accession code PRJNA701314.

The following dataset was generated:

| Author(s) | Year | Dataset title | Dataset URL | Database and Identifier |
|---|---|---|---|---|
| Hoermann A | 2021 | Amplicon-sequencing over the Anopheles gambiae CP, AP2 and Aper1 loci after non-autonomous gene drive | https://www.ncbi.nlm.nih.gov/Traces/study/?acc=PRJNA701314 | NCBI Sequence Read Archive, PRJNA701314 |

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
