## [Decision Letter]

**Acceptance summary:**

Engineered gene drives could potentially be used to spread genes that prevent malaria transmission in mosquitoes. In this study, the authors develop a proof-of-principle of effector components that would be part of a proposed integral gene drive. Such drives differ from standard gene drives by separating the Cas9 and effector components at different loci, with each one having biased inheritance, a useful strategy if the Cas9 has a substantial fitness cost.

**Decision letter after peer review:**

Thank you for submitting your article "Converting endogenous genes of the malaria mosquito into simple non-autonomous gene drives for population replacement" for consideration by *eLife*. Your article has been reviewed by 3 peer reviewers, and the evaluation has been overseen by a Reviewing Editor and Diethard Tautz as the Senior Editor. The following individuals involved in review of your submission have agreed to reveal their identity: Mara Lawniczak (Reviewer #2); Eric Marois (Reviewer #3).

The reviewers have discussed the reviews with one another and the Reviewing Editor has drafted this decision to help you prepare a revised submission.

Summary:

This paper demonstrates a number of important steps necessary for implementing the recently proposed "integral gene drive" strategy. In this approach, endogenous mosquito genes are hijacked to express a heterologous effector peptide intended to render mosquitoes resistant to human pathogens. Such drives differ from standard gene drives by separating the Cas9 and effector components at different loci, with each one having biased inheritance. This could be useful if the Cas9 has a substantial fitness cost and could also more easily target conserved sites of important genes compared to a standard drive. While it remains to be seen how effective this approach will be in practice, the paper provides valuable insights into how such gene drives could work in mosquitoes.

Essential revisions:

All reviewers found that overall the manuscript is technically sound. However, it was also felt that some of the stated conclusions were not yet fully supported by the performed analyses. After discussing these issues, we identified the following four major areas that we believe would need to be addressed in a revision in order to make this paper suitable for *eLife*:

1. Better quantification of target gene expression for AP2 and CP. To demonstrate the potential usefulness of the integral gene drive concepts, more evidence should be provided that the method can indeed reliably insert an intron and genes (at least gRNAs) without significantly reducing the expression of the target gene. So far, a quantitative technique was only used on one of the three targets (Aper1) and showed substantially reduced expression. For the two other targets only Westerns blots are presented. As pointed out in the reviews, these would need additional analysis and controls to be truly quantitative. If this cannot be shown, the authors would have to state this caveat very prominently in the manuscript, as it could severely limit the applicability of the integral gene drive concept.

2. Improved fitness assessment. Reviewers 3 and 4 point towards potential issues of the fitness analysis due to the fact that the lines were not outcrossed before they were interbred. We suggest that these experiments are redone with better controls. If this cannot be done, the caveats and limits would need to be clearly presented.

3. Additional controls for transmission blocking assays are needed if the authors want to sell Scorpine as a promising antimalarial effector. As detailed by Reviewers 2 and 3, the authors should consider adding inbred controls for phenotypic assays relating to transmission. They note themselves that the process of inbreeding can influence infections. It would thus seem important to try to mitigate this noise with replicate transgenic lines from independent G0s and potentially also "control" transgenic lines that have been through inbreeding, or even just several other inbred lines from the G3 background. Reviewer 3 additionally raised questions about the statistical analyses shown in Figure 5, which should be addressed.

4. Expanded treatment of resistance potential. It's unclear how exactly resistance alleles would be dealt with in the integral gene drive framework. At the very least, the authors should significantly expand the discussion of this issue. As pointed out by Reviewer 4, it is also not clear on what evidence the author's conclusion was based "that no end-joining was detected". This should be better explained.

Reviewer #2:

This is a compelling demonstration of a number of important steps that take population replacement gene drive for malaria control closer to reality. I have no major concerns and think the manuscript shows the authors have made substantial progress in (a) taking Integral Gene Drive (which is a recent idea from senior author Windbichler) into mosquitoes, (b) successfully removing marker genes to make the whole system more effective, (c) demonstrating that the approach works to express a molecule to reduce parasite infection rates in the lab while also making it possible to test these effector molecules in natural settings without risk of accidental drive release, and (d) also showing that drive is successful. I think the study is high impact.

Reviewer #3:

Hoermann et al. present a new gene engineering concept for disease vector mosquitoes, whereby endogenous mosquito genes are hijacked to express a heterologous effector peptide intended to render mosquitoes resistant to human pathogens. In addition, a synthetic intron added within the effector-coding sequence will express gRNAs for the CRISPR-Cas9 system, recognizing the transgene's own wild-type insertion locus. In the presence of a source of Cas9, the effector gene is thus able to home into a wild-type chromosome, triggering a gene drive effect that can increase the frequency of the modification in the mosquito population. A fluorescent marker, also cloned within the intron, is used at early steps to track the transgene, but is subsequently removed by Cre/lox excision to restore host gene + effector expression and to result in minimal genetic modification.

This is an extremely elegant procedure and a remarkable technical achievement, especially in such a difficult species as Anopheles gambiae. The choice of midgut-specific promoters to express anti-malaria effectors makes sense to target early stages of developement of parasites, before they had a chance to amplify in the mosquito. Using endogenous regulatory sequences without a need for promoter cloning alleviates the tedious work of individual promoter characterization. The molecular designs are well described, and the results likely to have a large future impact in the development of vector control tools, notwithstanding some weakness in assessing the antiparasitic effect of Scorpine in the transgenic mosquitoes (see below). I agree that this type of transgene should facilitate semi-field or field testing of candidate anti-parasitic effectors, before any true gene drive intervention is envisaged. I fully support publication of this manuscript in *eLife*, once the following major point concerning Figure 5is processed by the authors.

*P. falciparum* transmission blocking assays – Figure 5

I have several questions about figure 5.

– Are mosquitoes with 0 parasite taken into account in the calculation of the mean and median? This should be precise in the legend or in Exp procedures.

– Several replicates have been pooled to generate the figure, for each transgenic strain. Is this legitimate? i.e. were the mean oocyst number and prevalence, reflecting the quality of each ookinete culture, similar enough between replicates to allow pooling? If not, it would be more legitimate to show the result of a single representative replicate. Please provide a table with the raw parasite counts of the separate replicates in a Supplementary file so that readers can better judge these results. I note that panel C is very useful.

– I find the bar graph hard to interpret. The median M is represented either as a stroke inside some bars, or overlapping the x axis when M=0. The size of the bar doesn't represent the mean, m. Does it represent a confidence interval? this must be explained in the legend.

Maybe a dot plot where each dot represents the parasite counts of one mosquito would better represent these results?

– From my point of view, mosquito numbers in some of these infections may be too low to yield solid results. Especially in the ScoG-AP2 experiment: 37 mosquitoes in the G3 control with a prevalence of 51% means that only 19 mosquitoes across R=2 replicates contained parasites. This low number is associated with a risk of atypical outliers in the parasite counts, even if the statistical tests presented here show good significance. In the panel C analysis of these values, we see from the size of the squares that the replicate that had the highest statistical significance also had the smallest number of mosquitoes. The replicate with a larger N has only one *. For the Aper1-Sco line, N is large and the statistical significance is high (although panel C shows that one of the 4 replicates showed no difference) but I'm still somewhat unconvinced of the effect of scorpine in this line: the mean only drops from 10 to 6 parasites, prevalence drops from 37 to 21%. Combining this moderate effect with the facts that (1) some replicates sometimes show no Scorpine effect, (2) the Sco-CP line, which has a comparably high level of scorpine expression according to Suppl Figure. 3, shows the exact opposite, i.e. pro-parasitic effect, makes me doubt the antiparasitic effect of scorpine.

In the case of the ScoG-AP2 line, scorpine expression is only 1/10 to 1/8 of the expression in the other two lines, but seems to have a similar effect as in the highest (Aper1) expressing line: one possibility is that fusion to GFP stabilizes Scorpine so that lower expression results in higher activity, but a milder effect would have been logical if scorpine had a dose-dependent effect.

One caveat of these experiments is that the genetic background of the control mosquitoes (G3) is not exactly the same as the transgenics (G3 x KIL). There is a possibility that the KIL background contributed some alleles conferring elevated Plasmodium resistance (or the opposite in the case of Sco-CP). I would find the results more trustable if a control of equivalent genetic background could have been generated for each transgenic strain (in the process of homozygous line selection, the homozygous WT siblings could have been retained to serve as specific controls, though I know how demanding this work would have been…)

Another caveat is that we don't know the precise kinetics (e.g. between 0-36h post blood meal) of the scorpine protein midgut concentration in each transgenic line, and we don't know at what time point after the blood meal parasites would be most susceptible to killing by scorpine (probably between 3 and 24h, time after which they transform into protected cysts).

Taken together, the scorpine data is not highly conclusive and there remains much uncertainty about the efficacy of transgenically expressed Scorpine as an anti-plasmodium molecule. I'm not requesting additional experiments (though future long term assessments of these transgenic lines with new isogenic controls would be very interesting), but I invite the authors to downstate scorpine's potential effectiveness as an antimalarial effector in vivo. This does not decrease the importance of this work of which scorpine is only one aspect. A candidate molecule had to be chosen for these proof-of-principle experiments. Scorpine may not have been a very lucky choice, but its moderate (or opposite) effect should be seen as an interesting result in itself. The way is now open to test other possible candidates.

Reviewer #4:

Gene drives are alleles that bias their inheritance to spread through a population. Engineered gene drives could potentially be used to spread genes that prevent malaria transmission in mosquitoes. In this study, the authors develop a proof-of-principle of effector components that would be part of a proposed integral gene drives. Such drives different from standard gene drives by separating the Cas9 and effector components at different loci, with each one having biased inheritance, a useful strategy if the Cas9 has a substantial fitness cost (though it remains unclear if this is the case). They can also more easily target conserved sites of important genes compared to a standard drive, though this is not unique to the integral gene drive strategy. The Cas9 and effector components would be expressed from natural promoters, with introns and translation skipping utilized so that the original gene works properly and so gRNAs can be expressed within the intron. The authors showed that the effector component of such a drive performed as expected, and that both effectors and the target gene were expressed. Overall, the manuscript is a mostly sound technical demonstration of the effector component of an integral gene drive.

1. It's unclear how exactly resistance alleles would be dealt with in the author's strategy. While an integral gene drive could target an essential gene so that resistance alleles are nonviable, that doesn't seem to be the strategy here, since the authors needed to target a gene with a promoter that would be a good match for their effector. The need for both an essential gene and a suitable promoter in one package may thus limit the use of the integral gene drive strategy. Higher fitness costs associated with disruption of the gene may partially ameliorate this issue, but this was not confirmed in the current study (transgenic strains had lower fitness, but was this due to the drive, the effector, or the reduced expression of the host target gene?).

2. The authors removed their marker genes by surrounding them with LoxP sites and crossing their lines to Cre. This was justified since the authors believed that the presence of the marker would interfere with expression of the target gene, causing fitness issues. However, the authors found no sign of fitness reduction based on anecdotal (?) observations. Were these observations actually quantified, in which case they should be supplemental material? It could be particularly interesting in light of the fact that even without the marker, the transgenic strains suffered fitness effects. It would be nice if the decision to remove the marker was better justified in this section, based on the next section where it was found that the marker interfered with effector expression. Perhaps even combining or reversing the order of the sections would be appropriate (for example, consider first saying that the marker interferes with expression, then mention how this was expected and the marker could be removed, solving the problem).

3. Based on figure 3D-E, it appears that the target host gene has reduced expression even after the marker is removed. This is quite important for future considerations, yet seems to be glossed over. For example, if a target is chosen that can effectively help remove resistance alleles due to fitness costs from disrupting the target gene, this means that the gene drive will also suffer fitness costs.

4. The fitness analysis examining fecundity and hatch rates is not very informative. While similar fitness effects among the transgenic strains lends some weak evidence that inbreeding may account for the fitness reduction, variability between individuals certainly does not (after all, wild-type individuals were also highly variable). Also, if the Cre line has a different background than G3, wouldn't all the lines have received some of this background from prior crosses? Perhaps this could be the answer. It would nonetheless have been better for the authors to outcross the lines before inbreeding them, with similar inbreeding for the wild-type control, before doing this experiment. Because of the issues with this experiment, I'd suggest that it is conducted again with better controls or is moved to the supplement.

5. It's hard to believe that no end-joining took place, even though the last sentence of the results indicates that no end-joining was detected. Did the authors not sequence any progeny with the drive, to look for end-joining products formed from maternally deposited Cas9? Other studies with vasa-Cas9 in Anopheles saw this phenomenon occur at a high rate. For end-joining products formed as an alternative to HDR, was it 21 individuals that were sequenced (nine with Aper1 and twelve form the full AP2 sequencing)?

[Editors' note: further revisions were suggested prior to acceptance, as described below.]

Thank you for resubmitting your work entitled "Converting endogenous genes of the malaria mosquito into simple non-autonomous gene drives for population replacement" for further consideration by *eLife*. Your revised article has been reviewed by two peer reviewers and the evaluation has been overseen by Diethard Tautz as the Senior Editor and a Reviewing Editor.

Summary:

Engineered gene drives could potentially be used to spread genes that prevent malaria transmission in mosquitoes. In this study, the authors develop a proof-of-principle of effector components that would be part of a proposed integral gene drive. Such drives differ from standard gene drives by separating the Cas9 and effector components at different loci, with each one having biased inheritance, a useful strategy if the Cas9 has a substantial fitness cost. They might also be able to more easily target conserved sites of important genes compared to other drive designs. The Cas9 and effector components would be expressed from natural promoters, with introns and translation skipping utilized so that the original gene works properly and so gRNAs can be expressed within the intron. The authors showed that the effector component of such a drive performed as expected, and that both effectors and the target gene were expressed.

The manuscript has been substantially improved but there are still a few remaining issues raised by reviewer #4. We will be happy to accept the manuscript when these points have been addressed, which should only require a few short and easy edits to the text.

Reviewer #2:

The authors have addressed the comments from the first review well and I have no further concerns. I believe this should be published at *eLife*.

Reviewer #4:

I was reviewer #4 from the previous round. The authors made a good effort to improve their manuscript, and the extra experiments were certainly quite helpful. Some of the issues raised in the first review were successfully addressed. However, based on the data, I'm still not convinced by this manuscript that integral gene drives would offer a substantial advantage over other simpler systems or even that they can reach the capability of other rescue drives without a relatively greater development effort due to complications with target gene selection and expression (again, it is still certainly a possibility, but seems that future studies would really be needed to show this).

On a better note, only a few more revisions are needed in the current manuscript for all conclusions in to be sound and clearly presented. These necessary revisions can probably be done by changing the text without any new data or analysis (the length of my comments here is just an attempt to clarify what I'm suggesting – actual responses and revisions will almost certainly be far shorter).

On the assessment and considering of resistance alleles:

Regarding the new treatment of resistance, the authors say in their discussion that:

"Further studies e.g. in population cages could reveal to what degree the observed classes of mutations would be actively selected against, which would favour faithful transmission of the drive and is one predicted benefit of integral gene drives (33)."

This seems a bit misleading, since in the integral gene drive setup, such mutations could have a lower fitness cost than the drive, potentially strongly selecting against the drive once the pool of wild-type alleles is extinguished. The integral drive system, if anything, seems less able to deal with this sort of resistance allele than other designs unless it targets an essential gene, which the authors have noted does not appear to be the case with their designs in this study. This should be more clearly laid out at this point in the discussion.

Additionally, the amplicon sequencing was perhaps not the optimal experiment to directly measure the rate of paternal carryover. While it is certainly clear from the data that there was substantial maternal carryover, it can't be assessed quantitatively, and it is unclear if there was paternal carryover at all. This is because somatic Cas9 cleavage could account for some altered sequences. Based on the crossed, nearly half the offspring that were sequenced should have both integral drive alleles (almost 100% from homing) and vasa-Cas9 alleles (50% from normal inheritance). Somatic expression of Cas9 and gRNAs in these offspring could produce cleavage in individual cells, resulting in modified target site loci that would subsequently get detected during pooled amplicon sequenced. This needs to be clearly delineated in the results to explain the limitations of the experiment.

Side note: In the future, such an experiment could have been done with only offspring that did not inherit Cas9 (thus preventing acquisition of mutations from leaky somatic cleavage). To avoid amplicon sequencing, the authors could also have sequenced individual offspring (with or without Cas9) and looked for the presence of just 1-2 sequenced, which would confirm alleles by parental deposition (though only if Cas9 cleavage occurred in the very early embryo). The authors could also have done additional crosses with these offspring to isolate complete resistance allele sequences or assess the ability of drive and Cas9 heterozygotes the bias inheritance in a subsequent generation (lack of the ability to do so would be indicative of parental deposition forming resistance alleles).

For the resistance alleles that form in the germline, I'm now satisfied with the additional detail and interpretation. These seem to be below the rate that can be detected by the reasonable-sized experiment, which is quite nice.

On the fecundity/pupa hatch rate assessment:

The new data is quite informative and appears to support the notion that the differences in fecundity between lines is not based on the effector construct. There is no difference between the wild-type line and the GFP/CFP lines in Figure 4A. Yet, all the lines with GFP/CFP removed together with the Cre line have lower fecundity than the wild-type line in Figure 4B. Since removing the fluorescent protein should itself have no downside and probably even marginally improve fitness, this means that the reduced fitness is probably due to crossing with the Cre line. It might be best to spell this out a bit more thoroughly in the results. Additionally, the reduced fitness of the lines after crossing with Cre to remove the fluorescent proteins is now more of a minor point not really related to the integral drive effectors themselves, so perhaps this data should be moved to the supplement, leaving only the new data as a main figure.

For the pupa hatch rate, it should be noted if the relatively low level of significance can survive multiple-testing correction. This seems to be obliquely indicated in the text (that it cannot), but maybe something in the figure legend too so that readers don't focus too much on the "*".

On the other hand, the lack of fitness effects when the fluorescent proteins are present (and reducing the expression of the host gene as per Figure 3) does lend evidence to indicate that the target genes are perhaps not essential (as the authors note), increasing the difficulty in dealing with resistance, as noted above.

On the assessment of host gene expression changes:

The quantitative analysis provided in Figure 3D was certainly a useful addition, confirming the reduced expression of the host gene in two of three effectors, particularly in AP2. This indicates that the integral gene method may have difficulty when working with essential genes, which is likely needed to reduce the accumulation of resistance alleles. Yet, the relatively high expression of CP provides hope that the method could work for some target sites. Still, this may be a substantial issue that the integral drive strategy must overcome when targeting essential genes, and this is not clearly spelled out in the discussion when it certainly should be (next to the discussion of reduced host genes). After all, why bother restoring host gene expression in the first place if it unimportant? You could just insert the effector gene (scorpine or something else) without bothering to add the intron needed for host gene expression.

In line 384 (of the marked-up version) of the discussion, it is noted that the reduced mRNA could lead to normal protein levels. However, another possibility is that even somewhat reduced protein levels are not sufficient to have a fitness impact.

As a side note, the similar analysis of scorpine expression is also of note, in which one of the three constructs had very low expression. This further indicates that the integral gene drive method may not be suitable for all candidate sites. This is most likely true for more standard drive designs as well, though, so the issue here may not be due to the design of the integral design itself. However, the authors should still make this clearer in the Results section and in the discussion that one of the three systems had lower expression of scorpine. It is now only obliquely or indirectly referred to at present, but it needs to be clearly presented as a major result of the paper (even if it is also fine to indicate that the reduction was not likely due to the integral system itself).

---

## [Author Response]

Essential revisions:All reviewers found that overall the manuscript is technically sound. However, it was also felt that some of the stated conclusions were not yet fully supported by the performed analyses. After discussing these issues, we identified the following four major areas that we believe would need to be addressed in a revision in order to make this paper suitable for1. Better quantification of target gene expression for AP2 and CP. To demonstrate the potential usefulness of the integral gene drive concepts, more evidence should be provided that the method can indeed reliably insert an intron and genes (at least gRNAs) without significantly reducing the expression of the target gene. So far, a quantitative technique was only used on one of the three targets (Aper1) and showed substantially reduced expression. For the two other targets only Westerns blots are presented. As pointed out in the reviews, these would need additional analysis and controls to be truly quantitative. If this cannot be shown, the authors would have to state this caveat very prominently in the manuscript, as it could severely limit the applicability of the integral gene drive concept.

We now performed qPCR on all 6 lines for the endogenous host gene, as well as for the effector transgene (Figures 3C and D) allowing us to quantify expression better.

2. Improved fitness assessment. Reviewers 3 and 4 point towards potential issues of the fitness analysis due to the fact that the lines were not outcrossed before they were interbred. We suggest that these experiments are redone with better controls. If this cannot be done, the caveats and limits would need to be clearly presented.

The transgenic founders were outbred to G3 WT for 3 generations. We made this clearer in the Methods section. Once the fluorescent marker is removed, performing further rounds of outcrossing is too costly as it would involve molecular genotyping (there is no trackable visible marker). We now performed additional fitness assays with the lines including the marker-module (Figures 4A and C) as a more relevant comparator.

3. Additional controls for transmission blocking assays are needed if the authors want to sell Scorpine as a promising antimalarial effector. As detailed by Reviewers 2 and 3, the authors should consider adding inbred controls for phenotypic assays relating to transmission. They note themselves that the process of inbreeding can influence infections. It would thus seem important to try to mitigate this noise with replicate transgenic lines from independent G0s and potentially also "control" transgenic lines that have been through inbreeding, or even just several other inbred lines from the G3 background. Reviewer 3 additionally raised questions about the statistical analyses shown in Figure 5, which should be addressed.

We want to be clear that it is *not* our intention to sell Scorpine as an antimalarial effector in this paper, in fact it is neither mentioned in the title or abstract. The paper describes a new way of generating minimal non-autonomous gene drives and methods for the integration and co-expression of effectors from endogenous loci. If this was not clear, we have made an additional effort to make sure that message of the paper stays focussed and made changes to the results and Discussion sections.

We felt that for the sake of completeness it would nonetheless be necessary to report any obvious or extreme effects that the modifications of the target genes or the expression of Scorpine may have on transmission. The former in particular could have possibly had a strong effect in either direction. We chose Scorpine as a prototype effector, because it was previously reported to have an anti-malarial effect in paratransgenesis, although no transgenic Scorpine line was published before submission of this manuscript. As a reviewer mentions we needed to start somewhere.

We did observe a statistically significant effect on transmission in some of the experiments. Further experiments will be needed to tease apart the various effects that may contribute to this. The reviewers correctly identified inbreeding as a contributor, which is affecting most published homozygous transgenic lines to a greater or lesser degree (using inbreed controls is not a common procedure and could not settle this issue either). More generally, current methods to measure transmission blockage in mosquitoes have many known limitations some of which the reviewers also alluded to. A mixed rearing protocol is currently being established at Imperial and various molecular readouts (for various stages including sporozoites which we haven’t examined here) to deal with the shortfalls of the oocyte counting assay are being worked on by many labs. But dealing with all this here would go beyond the topic and scope of the present manuscript. We have changed the discussion to make sure it is clear why this experiment was done and what we can and cannot conclude from it.

Because these issues and pitfalls are well known, many gene drive papers with a focus on new drive designs for population replacement do not always conduct transmission experiments (for example https://www.pnas.org/content/early/2015/11/18/1521077112 with 700+ citations), perform their experiments without effectors or perform their experiments in *Drosophila*.

4. Expanded treatment of resistance potential. It's unclear how exactly resistance alleles would be dealt with in the integral gene drive framework. At the very least, the authors should significantly expand the discussion of this issue. As pointed out by Reviewer 4, it is also not clear on what evidence the author's conclusion was based "that no end-joining was detected". This should be better explained.

We now provided two sets of additional data for the current set of experiments with the Vasa-Cas9 driver. Dealing with another reviewer comment we performed two more biological replicates for the homing assays, and this allowed us to isolate more non-fluorescent individuals for Sanger sequencing. However, this is a numbers game, so for example with a 99.55% homing rate and a sample size of n=1980 we still could only find and analyse 9 non-fluorescent individuals for Aper1-Sco^GFP^ and this did not include any non-WT alleles. For CP two replicates showed 100% homing but we managed to obtain some non-fluorescent individuals in replicate 3.

To make our data more meaningful we now also performed amplicon sequencing of all pooled larvae that allows us to look at maternal deposition and germline modification quantitatively. Here we included also WT and the Cas9 strain as a background control to distinguish between pre-existing SNPs and newly emerged indels as a result of CRISPR/Cas9 activity.

The two datasets are looking at slightly different set of chromosomes (Figure 6—figure supplement 3A).

Reviewer #3:Hoermann et al. present a new gene engineering concept for disease vector mosquitoes, whereby endogenous mosquito genes are hijacked to express a heterologous effector peptide intended to render mosquitoes resistant to human pathogens. In addition, a synthetic intron added within the effector-coding sequence will express gRNAs for the CRISPR-Cas9 system, recognizing the transgene's own wild-type insertion locus. In the presence of a source of Cas9, the effector gene is thus able to home into a wild-type chromosome, triggering a gene drive effect that can increase the frequency of the modification in the mosquito population. A fluorescent marker, also cloned within the intron, is used at early steps to track the transgene, but is subsequently removed by Cre/lox excision to restore host gene + effector expression and to result in minimal genetic modification.This is an extremely elegant procedure and a remarkable technical achievement, especially in such a difficult species as Anopheles gambiae. The choice of midgut-specific promoters to express anti-malaria effectors makes sense to target early stages of developement of parasites, before they had a chance to amplify in the mosquito. Using endogenous regulatory sequences without a need for promoter cloning alleviates the tedious work of individual promoter characterization. The molecular designs are well described, and the results likely to have a large future impact in the development of vector control tools, notwithstanding some weakness in assessing the antiparasitic effect of Scorpine in the transgenic mosquitoes (see below). I agree that this type of transgene should facilitate semi-field or field testing of candidate anti-parasitic effectors, before any true gene drive intervention is envisaged. I fully support publication of this manuscript in eLife, once the following major point concerning Figure 5 is processed by the authors.*P. falciparum* transmission blocking assays – Figure 5I have several questions about figure 5.– Are mosquitoes with 0 parasite taken into account in the calculation of the mean and median? This should be precise in the legend or in Exp procedures.

Zeros were included when calculating mean and median and this was now clarified in the manuscript.

– Several replicates have been pooled to generate the figure, for each transgenic strain. Is this legitimate? i.e. were the mean oocyst number and prevalence, reflecting the quality of each ookinete culture, similar enough between replicates to allow pooling? If not, it would be more legitimate to show the result of a single representative replicate. Please provide a table with the raw parasite counts of the separate replicates in a Supplementary file so that readers can better judge these results. I note that panel C is very useful.

It is the common procedure in the field to pool replicates. We added Supplementary Table S6 with all the raw data of the transmission blocking assay.

– I find the bar graph hard to interpret. The median M is represented either as a stroke inside some bars, or overlapping the x axis when M=0. The size of the bar doesn't represent the mean, m. Does it represent a confidence interval? this must be explained in the legend.Maybe a dot plot where each dot represents the parasite counts of one mosquito would better represent these results?

We agree with the reviewer and converted Figure 5 to a dot blot.

– From my point of view, mosquito numbers in some of these infections may be too low to yield solid results. Especially in the ScoG-AP2 experiment: 37 mosquitoes in the G3 control with a prevalence of 51% means that only 19 mosquitoes across R=2 replicates contained parasites. This low number is associated with a risk of atypical outliers in the parasite counts, even if the statistical tests presented here show good significance. In the panel C analysis of these values, we see from the size of the squares that the replicate that had the highest statistical significance also had the smallest number of mosquitoes. The replicate with a larger N has only one *. For the Aper1-Sco line, N is large and the statistical significance is high (although panel C shows that one of the 4 replicates showed no difference) but I'm still somewhat unconvinced of the effect of scorpine in this line: the mean only drops from 10 to 6 parasites, prevalence drops from 37 to 21%. Combining this moderate effect with the facts that (1) some replicates sometimes show no Scorpine effect, (2) the Sco-CP line, which has a comparably high level of scorpine expression according to Suppl Figure. 3, shows the exact opposite, i.e. pro-parasitic effect, makes me doubt the antiparasitic effect of scorpine.

We thank the reviewer for looking into the details and for spotting a mistake in the ScoG-AP2 experiment, where the analysis of the G3 control only included 37 out of 78 available control mosquitoes. This was corrected.

In the case of the ScoG-AP2 line, scorpine expression is only 1/10 to 1/8 of the expression in the other two lines, but seems to have a similar effect as in the highest (Aper1) expressing line: one possibility is that fusion to GFP stabilizes Scorpine so that lower expression results in higher activity, but a milder effect would have been logical if scorpine had a dose-dependent effect.One caveat of these experiments is that the genetic background of the control mosquitoes (G3) is not exactly the same as the transgenics (G3 x KIL). There is a possibility that the KIL background contributed some alleles conferring elevated Plasmodium resistance (or the opposite in the case of Sco-CP). I would find the results more trustable if a control of equivalent genetic background could have been generated for each transgenic strain (in the process of homozygous line selection, the homozygous WT siblings could have been retained to serve as specific controls, though I know how demanding this work would have been…)Another caveat is that we don't know the precise kinetics (e.g. between 0-36h post blood meal) of the scorpine protein midgut concentration in each transgenic line, and we don't know at what time point after the blood meal parasites would be most susceptible to killing by scorpine (probably between 3 and 24h, time after which they transform into protected cysts).

We agree that one has to be cautious to draw a conclusion about the effect of Scorpine and this is also not the paper’s focus. Its mode of action might be locus-depended, and its efficacy might be dosage-dependent, which is not necessarily linear. The fact that we are exploring 3 quite different host genes for expression makes a direct comparison more difficult. Our intention was not to focus on Scorpine, but to explore different loci and design-options to express effector molecules within a novel gene drive framework. (See response to point 3 in essential revisions).

Taken together, the scorpine data is not highly conclusive and there remains much uncertainty about the efficacy of transgenically expressed Scorpine as an anti-plasmodium molecule. I'm not requesting additional experiments (though future long term assessments of these transgenic lines with new isogenic controls would be very interesting), but I invite the authors to downstate scorpine's potential effectiveness as an antimalarial effector in vivo. This does not decrease the importance of this work of which scorpine is only one aspect. A candidate molecule had to be chosen for these proof-of-principle experiments. Scorpine may not have been a very lucky choice, but its moderate (or opposite) effect should be seen as an interesting result in itself. The way is now open to test other possible candidates.

We agree and we have re-phrased several paragraphs to be more sober about Scorpine’s potential.

The reviewer is correct that we needed to start with a candidate effector and Scorpine was used as a proof-of-principle.

Reviewer #4:Gene drives are alleles that bias their inheritance to spread through a population. Engineered gene drives could potentially be used to spread genes that prevent malaria transmission in mosquitoes. In this study, the authors develop a proof-of-principle of effector components that would be part of a proposed integral gene drives. Such drives different from standard gene drives by separating the Cas9 and effector components at different loci, with each one having biased inheritance, a useful strategy if the Cas9 has a substantial fitness cost (though it remains unclear if this is the case). They can also more easily target conserved sites of important genes compared to a standard drive, though this is not unique to the integral gene drive strategy. The Cas9 and effector components would be expressed from natural promoters, with introns and translation skipping utilized so that the original gene works properly and so gRNAs can be expressed within the intron. The authors showed that the effector component of such a drive performed as expected, and that both effectors and the target gene were expressed. Overall, the manuscript is a mostly sound technical demonstration of the effector component of an integral gene drive.1. It's unclear how exactly resistance alleles would be dealt with in the author's strategy. While an integral gene drive could target an essential gene so that resistance alleles are nonviable, that doesn't seem to be the strategy here, since the authors needed to target a gene with a promoter that would be a good match for their effector. The need for both an essential gene and a suitable promoter in one package may thus limit the use of the integral gene drive strategy. Higher fitness costs associated with disruption of the gene may partially ameliorate this issue, but this was not confirmed in the current study (transgenic strains had lower fitness, but was this due to the drive, the effector, or the reduced expression of the host target gene?).

We did not know beforehand whether the host genes would turn out to be essential or not. To be more precise, we did not know whether their modification would lead to extreme fitness effects. We can also not rule out that their homozygous disruption (e.g. by NHEJ) does indeed have a fitness costs under either lab or field conditions and that such alleles would be selected against in a population. Generating clear knock-out alleles and studying their effect in population cages is a current undertaking. We made this distinction clearer in the discussion.

Essential genes are not absolutely required for this approach, although they provide additional benefits (this was discussed and modelled in the Nash et al. 2018 paper).

We now performed additional fitness assays with the lines including the marker-module (Figures 4A and C) as a more relevant comparator. These suggest that the modification of host gene expression can be ruled out as a source of major fitness effects.

2. The authors removed their marker genes by surrounding them with LoxP sites and crossing their lines to Cre. This was justified since the authors believed that the presence of the marker would interfere with expression of the target gene, causing fitness issues. However, the authors found no sign of fitness reduction based on anecdotal (?) observations. Were these observations actually quantified, in which case they should be supplemental material? It could be particularly interesting in light of the fact that even without the marker, the transgenic strains suffered fitness effects.

We now performed additional fitness assays with the lines including the marker-module (Figures 4A and C) as a more relevant comparator. These suggest that the modification of host gene expression can be ruled out as a source of major fitness effects.

It would be nice if the decision to remove the marker was better justified in this section, based on the next section where it was found that the marker interfered with effector expression. Perhaps even combining or reversing the order of the sections would be appropriate (for example, consider first saying that the marker interferes with expression, then mention how this was expected and the marker could be removed, solving the problem).

We tried to better justify this step in the text.

3. Based on figure 3D-E, it appears that the target host gene has reduced expression even after the marker is removed. This is quite important for future considerations, yet seems to be glossed over. For example, if a target is chosen that can effectively help remove resistance alleles due to fitness costs from disrupting the target gene, this means that the gene drive will also suffer fitness costs.

We now performed additional qPCR assays to quantify host gene mRNA expression. We improved the discussion to incorporate some of the considerations above. When combined with the new set of fitness experiments, again it appears that the modification of host gene expression can be ruled out as a source of major fitness effects at least in the assays we performed.

4. The fitness analysis examining fecundity and hatch rates is not very informative. While similar fitness effects among the transgenic strains lends some weak evidence that inbreeding may account for the fitness reduction, variability between individuals certainly does not (after all, wild-type individuals were also highly variable). Also, if the Cre line has a different background than G3, wouldn't all the lines have received some of this background from prior crosses? Perhaps this could be the answer. It would nonetheless have been better for the authors to outcross the lines before inbreeding them, with similar inbreeding for the wild-type control, before doing this experiment. Because of the issues with this experiment, I'd suggest that it is conducted again with better controls or is moved to the supplement.

The transgenic founders were outbred to G3 WT for 3 generations. Once the fluorescent marker is removed further rounds of outcrossing are too costly as it would involve molecular genotyping. Every published homozygous transgenic line is inbred to a certain degree, but it is not a common procedure to use inbreed wild-type controls as this ultimately can’t settle the question.

To complete this experiment and address some of the issues mentioned, we now performed additional fitness assays with the lines including the marker-module (Figures 4A and C) as a more relevant comparator. (see comments above).

5. It's hard to believe that no end-joining took place, even though the last sentence of the results indicates that no end-joining was detected. Did the authors not sequence any progeny with the drive, to look for end-joining products formed from maternally deposited Cas9? Other studies with vasa-Cas9 in Anopheles saw this phenomenon occur at a high rate. For end-joining products formed as an alternative to HDR, was it 21 individuals that were sequenced (nine with Aper1 and twelve form the full AP2 sequencing)?

See response to point 4 in essential revisions.

[Editors' note: further revisions were suggested prior to acceptance, as described below.]

[…] Reviewer #4:I was reviewer #4 from the previous round. The authors made a good effort to improve their manuscript, and the extra experiments were certainly quite helpful. Some of the issues raised in the first review were successfully addressed. However, based on the data, I'm still not convinced by this manuscript that integral gene drives would offer a substantial advantage over other simpler systems or even that they can reach the capability of other rescue drives without a relatively greater development effort due to complications with target gene selection and expression (again, it is still certainly a possibility, but seems that future studies would really be needed to show this).On a better note, only a few more revisions are needed in the current manuscript for all conclusions in to be sound and clearly presented. These necessary revisions can probably be done by changing the text without any new data or analysis (the length of my comments here is just an attempt to clarify what I'm suggesting – actual responses and revisions will almost certainly be far shorter).

We thank reviewer 4 for all the genuinely valuable comments, and the significant time he/she has invested in improving this manuscript. Fine-tuning of the integral gene drive designs is definitely required and further studies are already being undertaken.

On the assessment and considering of resistance alleles:Regarding the new treatment of resistance, the authors say in their discussion that:"Further studies e.g. in population cages could reveal to what degree the observed classes of mutations would be actively selected against, which would favour faithful transmission of the drive and is one predicted benefit of integral gene drives (33)."This seems a bit misleading, since in the integral gene drive setup, such mutations could have a lower fitness cost than the drive, potentially strongly selecting against the drive once the pool of wild-type alleles is extinguished. The integral drive system, if anything, seems less able to deal with this sort of resistance allele than other designs unless it targets an essential gene, which the authors have noted does not appear to be the case with their designs in this study. This should be more clearly laid out at this point in the discussion.

We modified this statement and also added a sentence to further improve this section making clear that while the targeted genes don’t appear to be essential (or rather, could readily be modified), we don’t know for sure that negative effects won’t result from their disruption.

Additionally, the amplicon sequencing was perhaps not the optimal experiment to directly measure the rate of paternal carryover. While it is certainly clear from the data that there was substantial maternal carryover, it can't be assessed quantitatively, and it is unclear if there was paternal carryover at all. This is because somatic Cas9 cleavage could account for some altered sequences. Based on the crossed, nearly half the offspring that were sequenced should have both integral drive alleles (almost 100% from homing) and vasa-Cas9 alleles (50% from normal inheritance). Somatic expression of Cas9 and gRNAs in these offspring could produce cleavage in individual cells, resulting in modified target site loci that would subsequently get detected during pooled amplicon sequenced. This needs to be clearly delineated in the results to explain the limitations of the experiment.Side note: In the future, such an experiment could have been done with only offspring that did not inherit Cas9 (thus preventing acquisition of mutations from leaky somatic cleavage). To avoid amplicon sequencing, the authors could also have sequenced individual offspring (with or without Cas9) and looked for the presence of just 1-2 sequenced, which would confirm alleles by parental deposition (though only if Cas9 cleavage occurred in the very early embryo). The authors could also have done additional crosses with these offspring to isolate complete resistance allele sequences or assess the ability of drive and Cas9 heterozygotes the bias inheritance in a subsequent generation (lack of the ability to do so would be indicative of parental deposition forming resistance alleles).

We included the possibility of somatic cleavage in the relevant section. This is an important point that got lost during revision and we thank the reviewer for picking up on it.

For the resistance alleles that form in the germline, I'm now satisfied with the additional detail and interpretation. These seem to be below the rate that can be detected by the reasonable-sized experiment, which is quite nice.On the fecundity/pupa hatch rate assessment:The new data is quite informative and appears to support the notion that the differences in fecundity between lines is not based on the effector construct. There is no difference between the wild-type line and the GFP/CFP lines in Figure 4A. Yet, all the lines with GFP/CFP removed together with the Cre line have lower fecundity than the wild-type line in Figure 4B. Since removing the fluorescent protein should itself have no downside and probably even marginally improve fitness, this means that the reduced fitness is probably due to crossing with the Cre line. It might be best to spell this out a bit more thoroughly in the results. Additionally, the reduced fitness of the lines after crossing with Cre to remove the fluorescent proteins is now more of a minor point not really related to the integral drive effectors themselves, so perhaps this data should be moved to the supplement, leaving only the new data as a main figure.

We now explain more clearly that inbreeding or the interbreeding steps involving the Cre line likely account for the observed fitness effects (and that the marker removal is not a likely contributor).

For the pupa hatch rate, it should be noted if the relatively low level of significance can survive multiple-testing correction. This seems to be obliquely indicated in the text (that it cannot), but maybe something in the figure legend too so that readers don't focus too much on the "*".

This is correct and depending on the test and correction performed (e.g. Kruskal-Wallis test with Dunn's correction) this comparison is not significant (panel D). However, if given the choice we’d rather not complicate the legend especially as we want to be conservative in reporting any possible fitness effects.

On the other hand, the lack of fitness effects when the fluorescent proteins are present (and reducing the expression of the host gene as per Figure 3) does lend evidence to indicate that the target genes are perhaps not essential (as the authors note), increasing the difficulty in dealing with resistance, as noted above.On the assessment of host gene expression changes:The quantitative analysis provided in Figure 3D was certainly a useful addition, confirming the reduced expression of the host gene in two of three effectors, particularly in AP2. This indicates that the integral gene method may have difficulty when working with essential genes, which is likely needed to reduce the accumulation of resistance alleles. Yet, the relatively high expression of CP provides hope that the method could work for some target sites. Still, this may be a substantial issue that the integral drive strategy must overcome when targeting essential genes, and this is not clearly spelled out in the discussion when it certainly should be (next to the discussion of reduced host genes). After all, why bother restoring host gene expression in the first place if it unimportant? You could just insert the effector gene (scorpine or something else) without bothering to add the intron needed for host gene expression.

We added a sentence to put essential genes and the necessity for sufficiently restored expression levels in context.

In line 384 (of the marked-up version) of the discussion, it is noted that the reduced mRNA could lead to normal protein levels. However, another possibility is that even somewhat reduced protein levels are not sufficient to have a fitness impact.

We rephrased the sentence to account for this possibility.

As a side note, the similar analysis of scorpine expression is also of note, in which one of the three constructs had very low expression. This further indicates that the integral gene drive method may not be suitable for all candidate sites. This is most likely true for more standard drive designs as well, though, so the issue here may not be due to the design of the integral design itself. However, the authors should still make this clearer in the Results section and in the discussion that one of the three systems had lower expression of scorpine. It is now only obliquely or indirectly referred to at present, but it needs to be clearly presented as a major result of the paper (even if it is also fine to indicate that the reduction was not likely due to the integral system itself).

We added a small section on the observed differences in expression strength to the discussion.